# DiaBlo: Diagonal Blocks Are Sufficient For Finetuning

**Selcuk Gurses**[1,*]   **Aozhong Zhang**[1,*]   **Yanxia Deng**[1]   **Xun Dong**[1]   **Xin Li**[1]
**Naigang Wang**[2]   **Penghang Yin**[1]   **Zi Yang**[1,†]
[1]University at Albany, SUNY   [2]IBM T. J. Watson Research Center
`{sgurses, azhang3, ydeng5, xdong5, xli48, pyin, zyang8}@albany.edu`
`nwang@us.ibm.com`

## Abstract

Fine-tuning is a critical step for adapting large language models (LLMs) to domain-specific downstream tasks. To mitigate the substantial computational and memory costs of full-model fine-tuning, Parameter-Efficient Fine-Tuning (PEFT) methods have been proposed to update only a small subset of model parameters. However, performance gaps between PEFT approaches and full-model fine-tuning still exist. In this work, we present *DiaBlo*, a simple yet effective PEFT approach that updates only the diagonal blocks of selected model weight matrices. Unlike Low-Rank Adaptation (LoRA) and its variants, DiaBlo eliminates the need for low-rank matrix products, thereby avoiding the reliance on auxiliary initialization schemes or customized optimization strategies to improve convergence. This design leads to stable and robust convergence while maintaining comparable memory efficiency and training speed to LoRA. Moreover, we provide theoretical guarantees showing that, under mild low-rank conditions, DiaBlo is more expressive than LoRA in the linear problem and converges to a stationary point of the general nonlinear full fine-tuning. Through extensive experiments across a range of tasks—including commonsense reasoning, arithmetic reasoning, code generation, and safety alignment—we show that fine-tuning only diagonal blocks is sufficient for strong and consistent performance. DiaBlo not only achieves competitive accuracy but also preserves high memory efficiency and fast fine-tuning speed. Codes are available at `https://github.com/ziyangjoy/DiaBlo`.

## 1 Introduction

Large language models (LLMs) (Achiam et al., 2023; Brown et al., 2020; Touvron et al., 2023a) have achieved remarkable success across a wide range of natural language processing tasks, including reasoning, generation, and alignment. These models, often consisting of billions of parameters, are typically pre-trained on broad, general-purpose corpora. However, to adapt them to specific downstream tasks or domains, fine-tuning is essential. Full fine-tuning, which updates all parameters of the model, has been shown to yield strong performance but is prohibitively expensive in terms of computational cost, memory usage, and storage—especially on resource-constrained devices.

To address these challenges, Parameter-Efficient Fine-Tuning (PEFT) methods (Ding et al., 2023; Han et al., 2024) have emerged as a promising alternative. These methods aim to retain the performance benefits of full fine-tuning while updating only a small fraction of the model's parameters. Early approaches, such as Prompt Tuning (Lester et al., 2021) and Prefix Tuning (Li & Liang, 2021), introduce task-specific embeddings or continuous prompts that steer model behavior without modifying its original weights. These methods are lightweight and straightforward to implement, making them suitable for low-resource adaptation scenarios. However, they often struggle with scalability and limited expressiveness, especially on complex tasks. To improve performance, more structured methods like Low-Rank Adaptation (LoRA) (Hayou et al., 2024; Hu et al., 2021) have been proposed. LoRA adds trainable low-rank matrices to the pretrained weight, significantly reducing the number of trainable parameters. It has demonstrated strong performance across a variety of applications.

---

* Equal contribution    † Corresponding author

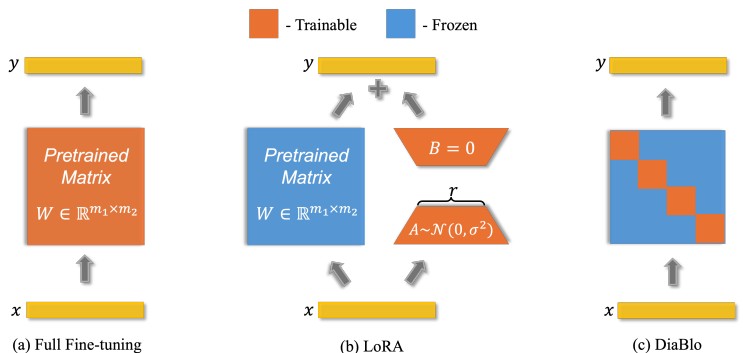

Figure 1: Comparison between full fine-tuning, LoRA, and proposed DiaBlo.

Despite its efficiency, LoRA and its variants often encounter issues such as unstable convergence and performance degradation. A growing number of work has proposed various extensions to LoRA, including tailored initialization schemes (Meng et al., 2024; Wang et al., 2024a;c) and structure-specific optimization strategies (Liu et al., 2024; Wang & Liang, 2024). Nonetheless, such improvements may lead to greater algorithmic complexity, potentially impacting implementation and efficiency. Sparsity-based methods, in contrast, aim to fine-tune only a subset of weight entries. Yet, most existing approaches rely on unstructured sparsity—through either random masking (Deng et al., 2024; Rios et al., 2025; Xu & Zhang, 2024) or importance-based selection (He et al., 2024)—which not only increases time complexity but also results in patterns that are difficult to exploit efficiently on modern hardware. $S^2FT$ (Yang et al., 2024a) is an efficient LLM adaptation framework that selectively updates sparse components while computing on dense submatrices. SparseLoRA (Khaki et al., 2025) employs a lightweight SVD-based estimator to identify sparse weights for loss and gradient computation, thereby accelerating LLM fine-tuning.

In this work, we propose DiaBlo, a simple yet effective PEFT framework that fine-tunes only the diagonal blocks of the model's weight matrices. Figure 1 illustrates the difference between full fine-tuning, LoRA, and DiaBlo. Unlike LoRA, which introduces low-rank adaptation through the product of two trainable matrices that often requires careful initialization and customized optimization to ensure stable training, DiaBlo directly updates a structured subset of the model's original weight parameters. By avoiding the use of matrix products, DiaBlo eliminates the inherent optimization difficulties associated with low-rank decomposition. This leads to greater training stability and more reliable convergence without the need for special tricks or tuning. Unlike existing fine-tuning approaches based on unstructured sparsity, DiaBlo's structured diagonal blocks preserve memory and computational efficiency while providing hardware-friendly patterns and enhanced representational flexibility. It integrates easily into standard training pipelines and consistently outperforms existing PEFT methods across a broad range of tasks, demonstrating that selectively updating diagonal blocks is a powerful and practical alternative for parameter-efficient fine-tuning.

As shown in Figure 2, DiaBlo clearly outperforms other PEFT baselines in both commonsense and arithmetic reasoning with the full-precision LLaMA2-7B model, and it maintains this advantage under quantized settings. Notably, DiaBlo demonstrates higher stability in 4-bit and 2-bit arithmetic reasoning tasks, where competing methods show significant performance degradation. These results highlight not only the expressive power of diagonal block updates but also the method's robustness across diverse domains. DiaBlo demonstrates strong convergence behavior, efficient resource usage, and broad applicability, making it a compelling alternative to existing parameter-efficient methods.

Our contributions are summarized as follows:

- **Sufficiency of Diagonal Block**: We propose DiaBlo, which fine-tunes only the diagonal blocks of model weight matrices. Through extensive experiments on tasks including commonsense reasoning, arithmetic reasoning, code generation, and safety alignment, we demonstrate that DiaBlo consistently achieves strong downstream performance, often surpassing LoRA and its variants in accuracy.

- **Theoretical Guarantees**: We provide theoretical justifications for DiaBlo. We show that, under mild *low-rank* assumptions on activations and gradients that often approximately hold in practice, DiaBlo converges to a stationary solution of full fine tuning (FT), provided the

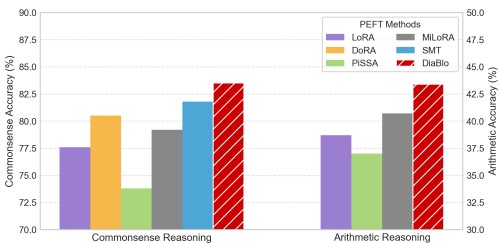 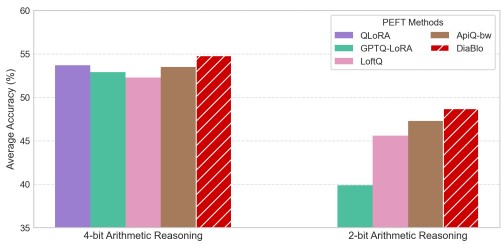

(a) Finetuning full-precision LLaMA2-7B          (b) Finetuning quantized LLaMA2-7B

Figure 2: Comparison of DiaBlo with other PEFT methods on finetuning LLaMA2-7B.

diagonal block size is sufficiently large. In the linear least squares setting, we further prove that DiaBlo converges to a global minimizer of the full FT objective and is strictly more expressive than LoRA under the same parameter budget.

- **Simple Optimization**: DiaBlo directly updates diagonal blocks using standard training pipelines, without relying on matrix product structures or specialized initialization, resulting in more stable and reliable fine-tuning.

- **High computation efficiency**: DiaBlo retains the low memory footprint and fast training of vanilla LoRA and other PEFT methods, ensuring practical usage for large fine-tuning.

## 2 BACKGROUND AND RELATED WORKS

Early PEFT techniques, such as Prompt Tuning (Han et al., 2021; Lester et al., 2021) and Prefix Tuning (Li & Liang, 2021), introduce learnable embeddings or token sequences that steer the model's behavior without modifying the core parameters. These methods are lightweight and easy to integrate, offering low computational overhead. While effective in low-resource or few-shot scenarios, prompt-based methods often struggle to match the performance of full fine-tuning on more complex tasks due to their limited expressiveness and adaptability. Adapter-based approaches (Houlsby et al., 2019; Rücklé et al., 2020; Wang et al., 2020) insert small, trainable bottleneck layers into the transformer architecture. These modules enable task-specific learning while keeping most of the model's parameters frozen. Variants of adapter tuning further improve performance through mechanisms such as attention fusion or parameter sharing. Although adapters may reach a good performance, they often require changes to model architecture and introduce additional inference-time latency. In addition, there exist decomposition-based approaches (Lingam et al., 2024; Liu et al., 2025; Wang et al., 2024e; Yang et al., 2024b;c; 2023) to reduce training or finetuning costs.

Low-Rank Adaptation (LoRA) (Hu et al., 2021) has emerged as one of the most popular PEFT methods. LoRA injects low-rank matrices into the update paths of existing model weights, significantly reducing the number of trainable parameters. Its effectiveness and simplicity have led to wide adoption, but its core reliance on the product of two low-rank matrices introduces optimization challenges. This structure can restrict representational flexibility and make training sensitive to initialization and optimization algorithms.

Several recent variants have been proposed to enhance the performance of LoRA. DoRA (Liu et al., 2024) introduces a novel optimization strategy that decouples the magnitude and direction of low-rank updates, leading to more stable training dynamics and better performance. Pissa (Meng et al., 2024) and MiLoRA (Wang et al., 2024a) focus on different initialization techniques for low-rank adapters. Pissa initializes the low-rank matrices using the largest singular values and singular vectors of weights, thereby preserving the most significant components of the weight matrix for faster convergence. On the other hand, MiLoRA takes a different approach by utilizing the smallest singular values during initialization. This strategy captures the finer details of the weight matrix. Additionally, LoRA-GA (Wang et al., 2024c) proposes to initialize low-rank matrices to align with gradients of full fine-tuning at the first step. While these LoRA-based methods have achieved notable improvements, they often introduce additional complexity in design and implementation. Furthermore, the dependence on matrix-product-based updates continues to present challenges for stable and efficient optimization. There exist many other variants of LoRA. We refer readers to (Kalajdzievski, 2023; Ponkshe et al., 2024; Valipour et al., 2022; Wang et al., 2024b; Zhang et al., 2023; 2025b; Zhong et al., 2024). QLoRA (Dettmers et al., 2023) reduces the memory footprint of fine-tuning by applying low-rank adaptation

on quantized models. Building on this idea, subsequent works (GPT, 2023; Deng et al., 2025; Li et al., 2023; Liao et al., 2024) design customized quantization algorithms and develop effective initialization strategies for the low-rank adaptation components, aiming to improve fine-tuning performance.

## 3 METHODOLOGY

### 3.1 DIABLO: FINETUNING DIAGONAL BLOCKS

**Motivation** Despite the success of LoRA-based fine-tuning methods, low-rank factorizations remain difficult to optimize in practice. In computational mathematics, problems such as low-rank approximation and low-rank matrix completion are well known to be non-convex and challenging (Ma et al., 2019; Tong et al., 2020; Tu et al., 2015). These difficulties are reflected in LoRA-based approaches, where the product of two trainable low-rank matrices can lead to unstable gradient flow and affect convergence. As a result, much recent work has focused on designing better initialization schemes and custom optimization strategies to stabilize training. The development of DiaBlo is motivated by these limitations. To this end, DiaBlo directly fine-tunes only the diagonal blocks of model weight matrices, avoiding matrix-product parameterizations entirely. This design simplifies the optimization landscape, improves training stability, and enables effective adaptation without introducing additional complexity.

**Algorithm** The state-of-the-art LLMs are constructed based on linear layers. Hence, it is sufficient to present our proposed approach on a simple linear layer. Consider a linear layer with input features $m_1$ and output features $m_2$,

$$\mathbf{Y} = \mathbf{XW},$$

where $\mathbf{Y} \in \mathbb{R}^{b \times m_2}, \mathbf{X} \in \mathbb{R}^{b \times m_1}, \mathbf{W} \in \mathbb{R}^{m_1 \times m_2}$, and $b$ is the batch size. For a given integer $N$ that is a factor of both $m_1, m_2$, we rewrite the weight $\mathbf{W}$ in the form of block matrices

$$\mathbf{W} = \begin{pmatrix} \mathbf{W}_{11} & \mathbf{W}_{12} & \cdots & \mathbf{W}_{1N} \\ \mathbf{W}_{21} & \mathbf{W}_{22} & \cdots & \mathbf{W}_{2N} \\ \vdots & \vdots & \ddots & \vdots \\ \mathbf{W}_{N1} & \mathbf{W}_{N2} & \cdots & \mathbf{W}_{NN} \end{pmatrix},$$

where $\mathbf{W}_{ij} \in \mathbb{R}^{d_1 \times d_2}, d_1 = \frac{m_1}{N}, d_2 = \frac{m_2}{N}$. During finetuning, only the diagonal blocks of $\mathbf{W}$, i.e., $\mathbf{W}_{11}, \ldots, \mathbf{W}_{NN}$, are trainable while all other blocks $\{\mathbf{W}_{ij}\}_{i \neq j}$ are frozen. DiaBlo also works when $N$ is not a common factor of $m_1, m_2$. In this case, we set $d_1 = \left\lceil \frac{m_1}{N} \right\rceil, d_2 = \left\lceil \frac{m_2}{N} \right\rceil$. At inference time, we pad $\mathbf{X}$ with zeros so that its dimension becomes $b \times N d_1$, perform the blockwise computation, and then truncate the resulting output back to the size $b \times m_2$. This situation is uncommon in practice because modern LLM architectures purposely choose hidden sizes and intermediate dimensions that are highly composite, often powers of two or products of small primes with powers of two, ensuring many convenient common factors. For instance, the Llama3-8B model has a hidden dimension of $4096 = 2^{12}$ and an FFN intermediate dimension of $14336 = 7 \times 2^{11}$. This allows us to choose $N = 2^k$ for $k = 1, \ldots, 11$. In practice, $N = 32, 64, 128$ are commonly used, as these choices are always common factors and align well with the trainable-parameter budgets of other PEFT methods.

**Efficient Implementation** For convenience of implementation, we use a similar adaptation method like LoRA and its variants. We introduce a block diagonal adaptation matrix $\mathbf{D}$ to the linear layer:

$$\mathbf{Y} = \mathbf{XW} = \mathbf{XW}_0 + \mathbf{XD}. \tag{1}$$

The adaptation matrix $\mathbf{D}$ is block diagonal

$$\mathbf{D} = \begin{pmatrix} \mathbf{D}_1 & \mathbf{0} & \cdots & \mathbf{0} \\ \mathbf{0} & \mathbf{D}_2 & \cdots & \mathbf{0} \\ \vdots & \vdots & \ddots & \vdots \\ \mathbf{0} & \mathbf{0} & \cdots & \mathbf{D}_N \end{pmatrix},$$

where $\mathbf{D}_i \in \mathbb{R}^{d_1 \times d_2}$. During fine-tuning, only $\mathbf{D}_i$'s are trainable. In real implementations, the adaptation matrix $\mathbf{D}$ can be saved as a tensor $\mathcal{D} \in \mathbb{R}^{N \times d_1 \times d_2}$, where $\mathcal{D}_{i,:,:} = \mathbf{D}_i$. During forward-

and back- propagation, it is not necessary to reconstruct the matrix $\mathbf{D}$ as well. The matrix product $\mathbf{XD}$ can be written as $(\mathbf{X}_1\mathbf{D}_1 \quad \cdots \quad \mathbf{X}_N\mathbf{D}_N)$, where $\mathbf{X}_i \in \mathbb{R}^{b \times d_1}$. Therefore, $\mathbf{XD}$ is mathematically equivalent to batched matrix multiplications, which can be computed efficiently on GPUs. In PyTorch, the operation can be easily implemented using `torch.einsum`. After reshaping $\mathbf{X}$ into tensor $\mathbf{X}$ of size $b \times N \times d_1$, $\mathbf{XD}$ can be computed as

$$\texttt{torch.einsum}(bNd_1, Nd_1d_2 \rightarrow bNd_2, \mathcal{X}, \mathcal{D}).$$

Let $\mathbf{g_Y}$ be the gradient of loss to the output $\mathbf{Y}$, then the gradient to $\mathbf{D}_i$ is

$$\mathbf{g}_{\mathbf{D}_i} = \mathbf{X}_i^\top \mathbf{g}_{\mathbf{Y}_i}, \tag{2}$$

where $\mathbf{g}_{\mathbf{Y}_i} = (\mathbf{g_Y})_{:,(i-1)d_2+1,id_2}$. Hence, the backpropagation of DiaBlo can be efficiently implemented using batched matrix multiplications as well.

**Initialization of DiaBlo**   LoRA and its variants typically represent adaptation as a low-rank product $\mathbf{AB}$. Standard initialization schemes set $\mathbf{A}$ using Kaiming uniform and $\mathbf{B}$ as zero, ensuring $\mathbf{AB} = \mathbf{0}$ while the gradients to $\mathbf{A}, \mathbf{B}$ do not vanish. However, such initialization often leads to suboptimal performance, as observed in recent studies (Meng et al., 2024; Wang et al., 2024a;c). Consequently, variants like Pissa (Meng et al., 2024), MiLoRA (Wang et al., 2024a), and LoRA-GA (Wang et al., 2024c) propose alternative initialization techniques to improve optimization behavior—e.g., Pissa uses large singular values, while MiLoRA leverages small singular values for initialization. In contrast, DiaBlo does not involve matrix products and can be initialized more straightforwardly. We simply initialize $\mathcal{D}$ as an **all-zero tensor**. This avoids issues related to vanishing gradients or entangled parameter updates, enabling stable and efficient training without the need for specialized initialization and optimization techniques.

## 3.2   Theoretical Analysis

**Comparison with LoRA**   The gradient with respect to the full weight matrix $\mathbf{W}$ is given by:

$$\mathbf{g_W} = \mathbf{X}^\top \mathbf{g}_Y = \begin{pmatrix} \mathbf{X}_1^\top \\ \vdots \\ \mathbf{X}_N^\top \end{pmatrix} \begin{pmatrix} \mathbf{g}_{Y_1} & \cdots & \mathbf{g}_{Y_N} \end{pmatrix} = \begin{pmatrix} \mathbf{X}_1^\top \mathbf{g}_{Y_1} & \cdots & \mathbf{X}_1^\top \mathbf{g}_{Y_N} \\ \vdots & \ddots & \vdots \\ \mathbf{X}_N^\top \mathbf{g}_{Y_1} & \cdots & \mathbf{X}_N^\top \mathbf{g}_{Y_N} \end{pmatrix}.$$

This implies that the gradient of the DiaBlo adaptation block $\mathbf{D}_i$ is exactly the gradient of the corresponding diagonal block $\mathbf{W}_{ii}$, i.e., $\mathbf{g}_{\mathbf{D}_i} = \mathbf{g}_{\mathbf{W}_{ii}}$. Therefore, optimizing DiaBlo's diagonal block adaptation is mathematically equivalent to fine-tuning the diagonal blocks of $\mathbf{W}$, which closely mirrors the behavior of full-model fine-tuning in those subspaces.

In contrast, for LoRA, where the adaptation takes the form $\mathbf{Y} = \mathbf{XW} + \mathbf{XAB}$, the gradients with respect to the low-rank matrices are computed as:

$$\mathbf{g_A} = \mathbf{g_W}\mathbf{B}^\top, \quad \mathbf{g_B} = \mathbf{A}^\top \mathbf{g_W}.$$

These gradients depend on the values of both $\mathbf{A}$ and $\mathbf{B}$, and can suffer from vanishing or unstable updates. As a result, LoRA's performance is sensitive to initialization strategies and often requires carefully designed optimization techniques. In contrast, DiaBlo directly applies gradients to the full-rank diagonal blocks, making it inherently more stable and effective for fine-tuning. The stability of DiaBlo is demonstrated by gradient norm comparisons with LoRA in Appendix A.3.4, where DiaBlo consistently exhibits lower variance, except for the vanishing $\mathbf{A}$-matrix gradients of LoRA in the early stage of fine-tuning.

**DiaBlo Converges to A Solution of Full FT**   In the linear least squares problem (LSQ) setting, DiaBlo is strictly more expressive than LoRA when the input matrix $\mathbf{X}$ has low rank. Consider the fine-tuning objective

$$\min_{\mathbf{W} \in \mathbb{R}^{m_1 \times m_2}} \frac{1}{2} \|\mathbf{Y} - \mathbf{XW}_0 - \mathbf{XW}\|_F^2, \tag{3}$$

where $\mathbf{X} \in \mathbb{R}^{b \times m_1}$, $\mathbf{Y} \in \mathbb{R}^{b \times m_2}$, and $\mathbf{W}_0 \in \mathbb{R}^{m_1 \times m_2}$. For DiaBlo, we instead optimize over block-diagonal updates:

$$\min_{\mathbf{D} = \mathrm{diag}(\mathbf{D}_1, \ldots, \mathbf{D}_N)} \frac{1}{2} \|\mathbf{Y} - \mathbf{XW}_0 - \mathbf{XD}\|_F^2, \tag{4}$$

where $\mathbf{D}_i \in \mathbb{R}^{d_1 \times d_2}$, $d_1 = \frac{m_1}{N}$, and $d_2 = \frac{m_2}{N}$. The following theorem shows that, under mild assumptions, any minimizer of the DiaBlo-LSQ (4) is also a minimizer of the full-LSQ (3).

**Theorem 1.** *Suppose that $\mathbf{X}$ is a generic rank-$r$ matrix. If the number of diagonal blocks $N \leq \frac{m_1}{r}$ is a common factor of $m_1, m_2$, then any solution to the DiaBlo-LSQ (4) also solves the full-LSQ (3).*

The proof of Theorem 1 is provided in Appendix A.1.1. According to the theorem, DiaBlo matches full FT performance with as few as $N d_1 d_2 = \frac{m_1 m_2}{N} \geq m_2 r$ trainable parameters. In contrast, LoRA must use a rank of at least $r$ to solve the full-LSQ, necessitating at least $(m_1 + m_2)r$ parameters. Consequently, DiaBlo remains strictly more expressive than LoRA under the same parameter constraints.

Remarkably, DiaBlo's effectiveness is not restricted to the linear setting. In general neural network fine-tuning, updating only diagonal blocks is often sufficient to match full FT performance. Our theoretical result substantiates this: DiaBlo converges to a stationary point of the full FT objective whenever the activation matrix $\mathbf{X}$ and output gradient $\mathbf{g_Y}$ exhibit a low-rank—a property empirically observed in recent literature (Chee et al., 2023; Zhang et al., 2024; Zhao et al., 2024). Formally:

**Theorem 2.** *Suppose the activation matrix $\mathbf{X} \in \mathbb{R}^{b \times m_1}$ and the linear output gradient $\mathbf{g_Y} \in \mathbb{R}^{b \times m_2}$ are generic with ranks $r_1$ and $r_2$, respectively. If $N$ is a common factor of $m_1, m_2$ and satisfies $N \geq \lceil \frac{r_1 N}{m_1} \rceil \lceil \frac{r_2 N}{m_2} \rceil$, then any adaptation $\mathbf{D} = \mathrm{diag}(\mathbf{D}_1, \ldots, \mathbf{D}_N) \in \mathbb{R}^{m_1 \times m_2}$ produced by DiaBlo that satisfies the stationarity conditions $\mathbf{g_{D_i}} = \mathbf{0}_{d_1 \times d_2}$ for all $1 \leq i \leq N$ yields adapted weights $\mathbf{W} = \mathbf{W}_0 + \mathbf{D}$ that is also a stationary point of the full FT objective, i.e., $\mathbf{g_W} = \mathbf{0}_{m_1 \times m_2}$.*

The proof is provided in Appendix A.1.2. Ignoring the ceiling operators in the assumption $N \geq \lceil \frac{r_1 N}{m_1} \rceil \lceil \frac{r_2 N}{m_2} \rceil$ gives an approximate condition $N \leq \frac{m_1 m_2}{r_1 r_2}$. Consequently, lower ranks for $\mathbf{X}$ and $\mathbf{g_Y}$ permit a larger numbers of blocks $N$ in Diablo. This reduces trainable parameters while maintaining the guarantee that the finetuned model is a stationary point of the full FT objective. An empirical validation of both the low-rank assumptions and the convergence of the full-weight gradient is presented in Appendix A.2.

## 4 EXPERIMENTS

In this section, we evaluate the performance of DiaBlo on a variety of benchmark datasets. Specifically, we assess its effectiveness on commonsense reasoning tasks, arithmetic reasoning tasks, code generation tasks, and safety alignment tasks. Our experiments are conducted using LLaMA2-7B (Touvron et al., 2023b), LLaMA3-8B (Dubey et al., 2024), and Mistral-7B-v0.1 (Jiang et al., 2023) models. All fine-tuning and evaluation are performed on a single NVIDIA A100 GPU with 80GB of memory. Additional experiments are shown in Appendix A.3 and the implementation details are provided in Appendix A.4.

### 4.1 COMMONSENSE REASONING

**Models and Datasets** Commonsense reasoning 170k includes eight benchmark datasets: BoolQ (Clark et al., 2019), PIQA (Bisk et al., 2020), SocialIQA (Sap et al., 2019), HellaSwag (Zellers et al., 2019), WinoGrande (Sakaguchi et al., 2021), ARC-Easy, ARC-Challenge (Clark et al., 2018), and OpenBookQA (Mihaylov et al., 2018). Each task is framed as a multiple-choice problem. We fine-tune the Llama2-7B, Llama3-8B, and Llama-13B models on the combined training data from all eight tasks, and report accuracy on the test set of each task individually.

**Results** The results on Commonsense reasoning tasks are shown in Table 1. We compare DiaBlo with existing methods, including DoRA (Liu et al., 2024), Pissa (Meng et al., 2024), MiLoRA (Wang et al., 2024a), and SMT (He et al., 2024). DiaBlo consistently outperforms existing PEFT baselines on commonsense reasoning tasks using Llama2-7B, Llama3-8B, and Llama-13B models in `FP16` precision. For Llama2-7B, DiaBlo achieves the highest average score of 83.5% while using only **0.52%** trainable parameters. For Llama3-8B, DiaBlo with $N = 64$ achieves the best overall average score of 87.3%. For the larger Llama-13B model, DiaBlo again demonstrates superior performance with a score of 84.9%, significantly outperforming strong baselines. The findings demonstrate DiaBlo is an efficient, scalable, and effective fine-tuning strategy.

**Comparison with SMT** SMT (He et al., 2024) proposes to select most important sub-matrices in pretrained weights based on magnitudes of gradients and then only fine-tune these sub-matrices.

Table 1: Commonsense reasoning results of fine-tuning Llama2-7B, Llama3-8B, and Llama-13B in `FP16`. Results marked with [†], [‡], [*] are taken from Dora (Liu et al., 2024), MiLoRA (Wang et al., 2024a), and SMT (He et al., 2024), respectively.

| PEFT | $r/N$ | #Params | BoolQ | PIQA | SIQA | HellaS | WinoG | ARC-e | ARC-c | OBQA | AVG |
|------|-------|---------|-------|------|------|--------|-------|-------|-------|------|-----|
| **Llama2-7B** | | | | | | | | | | | |
| Full FT | N/A | 100% | 73.3 | 85.7 | 81 | 90.2 | 86.9 | 88.6 | 77.4 | 85.2 | 83.5 |
| Zero-shot | N/A | 0% | 61.9 | 47.4 | 2.2 | 12.3 | 0.0 | 12.5 | 10.0 | 23.4 | 21.2 |
| LoRA[†] | $r=32$ | 0.83% | 69.8 | 79.9 | 79.5 | 83.6 | 82.6 | 79.8 | 64.7 | 81.0 | 77.6 |
| DoRA[†] | $r=32$ | 0.83% | 71.8 | 83.7 | 76.0 | 89.1 | 82.6 | 83.7 | 68.2 | 82.4 | 79.7 |
| DoRA[†] | $r=16$ | 0.42% | 72.0 | 83.1 | 79.9 | 89.1 | 83.0 | 84.5 | 71.0 | 81.2 | 80.5 |
| Pissa[‡] | $r=32$ | 0.83% | 67.6 | 78.1 | 78.4 | 76.6 | 78.0 | 75.8 | 60.2 | 75.6 | 73.8 |
| MiLoRA[‡] | $r=32$ | 0.83% | 67.6 | 83.8 | 80.1 | 88.2 | 82.0 | 82.8 | 68.8 | 80.6 | 79.2 |
| SMT[*] | N/A | 0.84% | 72.0 | 83.8 | 80.8 | 93.3 | 82.8 | 86.7 | 74.0 | 81.0 | 81.8 |
| SMT(Best)[*] | N/A | 4.91% | 72.6 | 85.2 | 82.0 | 94.4 | 85.7 | 87.8 | 74.5 | 85.0 | 83.4 |
| DiaBlo | $N=64$ | 1.04% | 73.9 | 84.8 | 81.7 | 90.0 | 85.0 | 87.9 | 76.8 | 86.8 | 83.4 |
| DiaBlo | $N=128$ | 0.52% | 74.8 | 85.5 | 80.9 | 89.9 | 85.7 | 88.5 | 76.3 | 86.0 | **83.5** |
| **Llama3-8B** | | | | | | | | | | | |
| Full FT | N/A | 100% | 76.4 | 89.7 | 82.5 | 95.5 | 89.6 | 92.9 | 84.3 | 89.2 | 87.5 |
| Zero-shot | N/A | 0% | 56.3 | 66.3 | 32.2 | 25.2 | 26.6 | 24.1 | 21.5 | 24.8 | 34.6 |
| LoRA[†] | $r=32$ | 0.78% | 70.8 | 85.2 | 79.9 | 91.7 | 84.3 | 84.2 | 71.2 | 79.0 | 80.8 |
| DoRA[†] | $r=32$ | 0.78% | 74.6 | 89.3 | 79.9 | 95.5 | 85.6 | 90.5 | 80.4 | 85.8 | 85.2 |
| DoRA[†] | $r=16$ | 0.39% | 74.5 | 88.8 | 80.3 | 95.5 | 84.7 | 90.1 | 79.1 | 87.2 | 85.0 |
| Pissa[‡] | $r=32$ | 0.78% | 67.1 | 81.1 | 77.2 | 83.6 | 78.9 | 77.7 | 63.2 | 74.6 | 75.4 |
| MiLoRA[‡] | $r=32$ | 0.78% | 68.8 | 86.7 | 77.2 | 92.9 | 85.6 | 86.8 | 75.5 | 81.8 | 81.9 |
| SMT[*] | N/A | 0.71% | 75.7 | 88.4 | 81.4 | 96.2 | 88.2 | 92.7 | 83.2 | 88.6 | 86.8 |
| SMT(best)[*] | N/A | 3.01% | 75.1 | 89.9 | 82.4 | 96.3 | 88.8 | 92.6 | 82.8 | 89.6 | 87.2 |
| DiaBlo | $N=64$ | 1.04% | 75.8 | 90.8 | 80.7 | 95.6 | 89.9 | 93.4 | 83.0 | 89.2 | **87.3** |
| DiaBlo | $N=128$ | 0.52% | 76.1 | 90.3 | 81.5 | 95.7 | 89.7 | 93.4 | 83.4 | 87.2 | 87.2 |
| **Llama-13B** | | | | | | | | | | | |
| Full FT | N/A | 100% | 74.8 | 86.2 | 80.5 | 90.5 | 87.1 | 88.6 | 75.4 | 88.0 | 83.9 |
| Zero-shot | N/A | 0% | 59.5 | 47.4 | 16.0 | 22.9 | 0.3 | 15.8 | 12.0 | 24.0 | 24.7 |
| LoRA | $r=32$ | 0.67% | 72.1 | 83.5 | 80.5 | 90.5 | 83.7 | 82.8 | 68.3 | 82.4 | 80.5 |
| DoRA | $r=32$ | 0.68% | 72.5 | 85.3 | 79.9 | 90.1 | 82.9 | 82.7 | 69.7 | 83.6 | 80.8 |
| DiaBlo | $N=64$ | 1.06% | 74.7 | 86.9 | 81.2 | 91.5 | 87.4 | 89.8 | 78.2 | 90.2 | **84.9** |
| DiaBlo | $N=128$ | 0.53% | 75.7 | 84.8 | 82.0 | 90.6 | 86.8 | 90.1 | 77.1 | 89.2 | 84.5 |

In contrast, DiaBlo directly selects diagonal blocks from the pretrained weights, offering a simpler and more implementation-friendly approach. The experiments further demonstrate that fine-tuning diagonal blocks provide even better results. As shown in Table 1, DiaBlo achieves an average score of 83.5% on Llama2-7B, significantly outperforming SMT's 81.8% while using a less number of trainable parameters. Even when SMT increases its trainable parameter to 4.91% to achieve its best result, DiaBlo, with only 0.52% trainable parameters, still attains better performance. A similar trend holds for Llama3-8B, where DiaBlo outperforms SMT even when SMT uses considerably more trainable parameters. These results demonstrate that fine-tuning only the diagonal blocks of pretrained weights is not only highly efficient but also sufficient to achieve strong performance.

## 4.2 ARITHMETIC REASONING

**Models and Datasets** We evaluate DiaBlo on arithmetic reasoning using the MetaMathQA (Yu et al., 2023) dataset, which consists of 395K samples augmented from the training sets of GSM8K (Cobbe et al., 2021) and MATH (Hendrycks et al., 2021). We fine-tune the LLaMA2-7B model on this dataset and evaluate performance on the test sets of GSM8K and MATH. The training follows the setting in MiLoRA and we report results from the final checkpoint and use the Exact Match (EM) ratio to measure accuracy.

**Results** As shown in Table 2, on arithmetic reasoning tasks, DiaBlo demonstrates strong performance, achieving results that match or exceed full fine-tuning while maintaining parameter efficiency. With $N = 32$ blocks and only 2.09% trainable parameters, DiaBlo attains an average accuracy of 43.4%, slightly outperforming full fine-tuning with 43.2% and significantly surpassing LoRA,

Table 2: Performance of fine-tuning Llama2-7B on arithmetic reasoning tasks. Results of baseline methods are taken from MiLoRA (Wang et al., 2024a).

| Method | r/N | #Params | GSM8K | MATH | AVG |
|---|---|---|---|---|---|
| Full FT | N/A | 100% | **66.5** | 19.8 | 43.2 |
| Zero-shot | N/A | 0% | 2.2 | 0.0 | 1.1 |
| LoRA | $r = 64$ | 1.67% | 60.6 | 16.9 | 38.7 |
| PiSSA | $r = 64$ | 1.67% | 58.2 | 15.8 | 37.0 |
| MiLoRA | $r = 64$ | 1.67% | 63.5 | 17.8 | 40.7 |
| DiaBlo | $N = 32$ | 2.09% | 66.3 | **20.4** | **43.4** |
| DiaBlo | $N = 64$ | 1.04% | **66.5** | 17.6 | 42.1 |

Table 3: Performance on code generation and safety alignment tasks. Results of baseline methods are taken from LoRI (Zhang et al., 2025a).

| Model | Method | r/N | #Params | HumanEval Pass@1 | Pass@5 | Pass@10 | HEx-PHI |
|---|---|---|---|---|---|---|---|
| **Llama3-8B** | Zero-shot | N/A | 0% | 28.5 | 39.8 | 45.2 | 78.7 |
| | LoRA | $r = 32$ | 1.12% | 34.7 | 46.4 | 50.8 | 91.6 |
| | DoRA | $r = 32$ | 1.12% | 33.1 | 44.0 | 48.6 | 93.6 |
| | LoRI | $r = 32$ | 0.56% | **43.2** | **57.6** | 63.2 | 92.8 |
| | DiaBlo | $N = 64$ | 1.51% | **43.2** | 57.4 | **63.5** | **97.6** |
| | DiaBlo | $N = 128$ | 0.76% | 39.4 | 55.0 | 61.9 | 96.3 |
| **Mistral-7B** | Zero-shot | N/A | 0% | 28.6 | 38.2 | 41.9 | 80.5 |
| | LoRA | $r = 32$ | 1.25% | 33.8 | 42.4 | 45.3 | 91.9 |
| | DoRA | $r = 32$ | 1.25% | 33.7 | 42.6 | 46.8 | 95.3 |
| | LoRI | $r = 32$ | 0.63% | 33.8 | 42.0 | 45.1 | 94.7 |
| | DiaBlo | $N = 64$ | 1.68% | **34.4** | 44.8 | 48.7 | **98.8** |
| | DiaBlo | $N = 128$ | 0.84% | 34.0 | **45.7** | **49.1** | **98.8** |

PiSSA, and MiLoRA by **4.7%**, **6.4%**, and **2.7%**, respectively. Even with $N = 64$ blocks and 1.04% trainable parameters, DiaBlo with fewer trainable parameters maintains high performance with an average accuracy of 42.1%, demonstrating that it remains effective under restrictive parameter budgets. Notably, DiaBlo achieves the highest score of **20.4%** on the MATH dataset among all methods, including full fine-tuning.

## 4.3 Code Generation and Safety Alignment

**Models and Datasets** We further evaluate the performance of DiaBlo on code generation and safety alignment tasks, following the training setting in LoRI (Zhang et al., 2025a). For code generation, we fine-tune on the CodeAlpaca dataset (Chaudhary, 2023) and evaluate using the HumanEval benchmark (Chen et al., 2021) with standard metrics: Pass@1, Pass@5, and Pass@10. For safety alignment, we fine-tune on the SaferPaca dataset (Bianchi et al., 2023), an extension of Alpaca-Cleaned (Taori et al., 2023), and assess performance based on the refusal rate to harmful prompts from the HEx-PHI dataset (Qi et al., 2023). We conduct experiments on both the LLaMA3-8B and Mistral-7B models.

**Results** The experimental results are summarized in Table 3. DiaBlo achieves strong performance in the code generation task HumanEval on Llama3-8B. With 1.51% trainable parameters, DiaBlo reaches a Pass@1 score of 43.2% and a Pass@10 score of 63.5%, outperforming all other methods. Its Pass@5 score is also very cloes to the best result obtained by LoRI. With a reduced parameter budget of 0.76%, DiaBlo still maintains high performance, achieving a Pass@10 score of 61.9%. On Mistral-7B, DiaBlo with $N = 128$ achieves the best Pass@5 and Pass@10 scores and the second-best Pass@1 score, while DiaBlo with $N = 64$ achieves the best Pass@1 score and the second-best Pass@5 and Pass@10 scores, outperforming LoRA, DoRA, and LoRI. In terms of HEx-PHI, DiaBlo outperforms all other methods with scores of 97.6% on Llama3-8B and 98.8% on Mistral-7B. These findings highlight DiaBlo's effectiveness in both code generation and safety alignment, while maintaining a high level of parameter efficiency.

## 4.4 Fine-tuning with Quantized Models

To further reduce the memory footprint of fine-tuning, QLoRA (Guo et al., 2024) and its variants store the frozen pretrained weights in low-precision while training small low-rank adaptation modules in

Table 4: Performance of fine-tuning quantized Llama2-7B and Llama2-13B models on arithmetic reasoning tasks. Results of baseline methods are taken from ApiQ (Liao et al., 2024).

| | Method | r/N | #Params | GSM8K | SVAMP | MAWPS | AQuA | AVG |
|---|---|---|---|---|---|---|---|---|
| | **Llama2-7B** | | | | | | | |
| **4-bit** | QLoRA | $r=64$ | 112M | 42.7 | 58.7 | 87.3 | 26.4 | 53.7 |
| | GPTQ-LoRA | $r=64$ | 112M | 43.0 | 58.4 | 86.1 | 24.3 | 52.9 |
| | LoftQ | $r=64$ | 112M | 41.7 | 56.0 | 86.3 | 25.3 | 52.3 |
| | ApiQ-bw | $r=64$ | 112M | 43.2 | 59.0 | 85.7 | 26.0 | 53.5 |
| | MagR-DiaBlo | $N=64$ | 70M | 44.1 | 59.0 | 89.5 | 26.4 | **54.8** |
| **2-bit** | QLoRA | $r=64$ | 112M | 0.9 | 1.5 | 0.8 | 5.1 | 2.1 |
| | GPTQ-LoRA | $r=64$ | 112M | 21.7 | 39.0 | 76.6 | 22.1 | 39.9 |
| | MagR-LoRA | $r=64$ | 112M | 30.9 | 46.9 | 86.6 | 20.5 | 46.2 |
| | LoftQ | $r=64$ | 112M | 29.5 | 45.8 | 83.6 | 23.2 | 45.6 |
| | ApiQ-bw | $r=64$ | 112M | 31.2 | 49.0 | 83.9 | 23.9 | 47.3 |
| | GPTQ-DiaBlo | $N=64$ | 70M | 33.3 | 50.9 | 84.0 | 22.8 | 47.8 |
| | MagR-DiaBlo | $N=64$ | 70M | 32.1 | 51.5 | 87.0 | 24.0 | **48.7** |
| | **Llama2-13B** | | | | | | | |
| **4-bit** | QLoRA | $r=64$ | 239M | 54.8 | 69.4 | 87.0 | 26.8 | 59.5 |
| | GPTQ-LoRA | $r=64$ | 239M | 53.2 | 67.5 | 85.3 | 25.6 | 57.9 |
| | LoftQ | $r=64$ | 239M | 54.9 | 66.5 | 87.7 | 23.9 | 58.3 |
| | ApiQ-bw | $r=64$ | 239M | 55.3 | 67.4 | 87.8 | 25.6 | 59.0 |
| | MagR-DiaBlo | $N=64$ | 189M | 55.5 | 66.9 | 89.9 | 26.0 | **59.6** |
| **2-bit** | QLoRA | $r=64$ | 239M | 0.5 | 0.7 | 0.1 | 0.9 | 0.6 |
| | GPTQ-LoRA | $r=64$ | 239M | 31.9 | 49.6 | 82.5 | 23.6 | 46.9 |
| | MagR-LoRA | $r=64$ | 239M | 38.6 | 60.5 | 81.1 | 24.8 | 51.2 |
| | LoftQ | $r=64$ | 239M | 37.0 | 55.9 | 87.7 | 21.7 | 50.6 |
| | ApiQ-bw | $r=64$ | 239M | 43.1 | 59.2 | 85.1 | 23.4 | 52.7 |
| | GPTQ-DiaBlo | $N=64$ | 189M | 41.7 | 58.0 | 87.4 | 23.2 | 52.6 |
| | MagR-DiaBlo | $N=64$ | 189M | 44.5 | 59.9 | 88.2 | 27.6 | **55.1** |

higher precision. In this section, we explore the compatibility of the proposed DiaBlo with quantized models and evaluate its performance.

**Models and Datasets**  We evaluate DiaBlo on quantized models. We fine-tune the quantized LLaMA2-7B and LLaMA2-13B models, which are quantized by GPTQ (Frantar et al., 2022) and MagR (Zhang et al., 2024). All models are fine-tuned on Math10K (Hu et al., 2023) and then evaluated on the test sets of AQuA (Ling et al., 2017), GSM8K (Cobbe et al., 2021), MAWPS (Koncel-Kedziorski et al., 2016), and SVAMP (Patel et al., 2021).

**Results**  We evaluate DiaBlo on 4-bit and 2-bit quantized Llama2-7B and Llama2-13B models. The results are presented in Table 4. For the Llama2-7B model, MagR-DiaBlo achieves the highest average accuracy of 54.8% in the 4-bit setting. In the more challenging 2-bit setting, it again outperforms existing baselines with an average accuracy of 48.7%. The results on the larger Llama2-13B model further demonstrate DiaBlo's scalability and robustness. Its advantage is particularly stark in the ultra-low 2-bit setting, where it achieves an average score of 55.1%, substantially outperforming baselines. The LoftQ and ApiQ methods jointly determine quantized weights and LoRA initialization to enhance fine-tuning performance. In contrast, DiaBlo directly fine-tunes quantized models without any customized quantization procedure or special initialization, highlighting its practical simplicity while still achieving superior or comparable results. To isolate the impact of quantization methods, we also evaluate GPTQ-DiaBlo and MagR-LoRA under the 2-bit setting. Under both GPTQ quantization and MagR quantization, simply replacing LoRA with DiaBlo leads to substantial performance improvements. This indicates that the observed gains come primarily from the fine-tuning method DiaBlo rather than from changes in the quantization method.

### 4.5 MEMORY AND COMPUTATION EFFICIENCY OF DIABLO

The simple structure of DiaBlo allows the use of efficient batched operations, thus reducing training time and memory usage. In this subsection, we evaluate the memory consumption and training speed of DiaBlo and compare them with other PEFT training methods.

Consider a linear layer $\mathbf{Y} = \mathbf{X}\mathbf{W}$, where $\mathbf{W} \in \mathbb{R}^{m \times m}$, $\mathbf{X} \in \mathbb{R}^{b \times m}$. Let $N$ be the number of blocks in DiaBlo. Each diagonal block is of size $d \times d$, where $d = \frac{m}{N}$, and the total number of trainable

Table 5: Comparison of DiaBlo with random sparse patterns on GSM8K. Left part reports accuracy from retaining sparse updates, and right part reports accuracy and training time from fine-tuning.

| Method | Retained Sparse Update Accuracy (%) | | | Fine-tuning (Sparsity 1/64) | |
|---|---|---|---|---|---|
| | Sparsity 1/32 | Sparsity 1/64 | Sparsity 1/128 | Accuracy (%) | Time (min) |
| DiaBlo | **67.30** | **65.47** | **63.11** | **67.68** | 17.26 |
| Random Entries | 65.98 | 64.05 | 62.19 | 65.35 | 26.51 |
| Random Block | 65.96 | 63.75 | 62.98 | 64.86 | 29.76 |
| Random Column | 65.73 | 62.93 | 61.66 | 65.19 | **17.01** |
| Random Row | 49.14 | 24.08 | 2.85 | 61.71 | 17.76 |

parameters is $Nd^2$. The forward pass using the block matrix structure requires operations $bNd^2$. In comparison, LoRA with rank $r$ has $2mr$ trainable parameters and requires $2bmr$ operations during the forward pass. When DiaBlo and LoRA are configured to have the same number of trainable parameters, that is, $Nd^2 = 2mr$, it is straightforward to show that the computational cost of the forward pass is also the same. A similar argument applies to the backward pass. Therefore, DiaBlo has the same theoretical computational complexity and memory footprint as LoRA when they share the same number of trainable parameters. This indicates that DiaBlo is as efficient as vanilla LoRA in terms of resource usage and computation complexity.

To further validate efficiency, we benchmark the fine-tuning speed of DiaBlo on the 170k commonsense reasoning dataset, comparing it against LoRA and DoRA. All experiments are conducted on a single A100 GPU with 80GB memory, using a batch size of 8 and a gradient accumulation step of 4. DiaBlo with $N = 64$ and $N = 128$ matches the training speed of LoRA with rank $r = 32$, requiring **170** minutes per epoch, whereas DoRA is substantially slower at 480 minutes per epoch. These results confirm the practical efficiency of the proposed DiaBlo method.

### 4.6 SUFFICIENCY OF DIAGONAL BLOCKS

We investigate whether diagonal blocks provide a more effective subset for fine-tuning compared to alternative sparsity patterns. Following the methodology of Deng et al. (2024), we first analyze the performance retained from a fully fine-tuned model. Specifically, we fully fine-tune LLaMA3-8B on GSM8K and compute the update matrix $\Delta = \mathbf{W}_{\text{finetuned}} - \mathbf{W}_{\text{pretrained}}$. We then measure accuracy after reapplying only selected parts of $\Delta$, scaled by a factor $s$, to the pre-trained weights. We compare DiaBlo's diagonal block selection with random entries, random blocks of the same size as DiaBlo, random columns, and random rows. As shown in Table 5, retaining diagonal blocks preserves accuracy substantially better than all other sparsity patterns, and degrades more gracefully as sparsity increases. We further evaluate these sparse patterns in direct fine-tuning. Table 5 shows that DiaBlo not only achieves the highest accuracy but also runs more efficiently than unstructured random methods, while avoiding the severe accuracy loss of row- or block-based updates. Together, these results demonstrate that diagonal blocks form a principled and hardware-friendly structure for sparse fine-tuning, sufficient to capture task-relevant updates.

### 5 CONCLUSION

In this work, we introduced DiaBlo, a simple yet effective parameter-efficient fine-tuning (PEFT) framework that directly updates diagonal blocks of model weight matrices. Unlike low-rank approaches such as LoRA and its many variants, DiaBlo avoids matrix-product-based adaptations, eliminating the need for special initialization or customized optimization techniques. Our theoretical guarantees show that, under mild low-rank conditions, DiaBlo converges to a stationary point of the general full fine-tuning and is more expressive than LoRA in the linear least squares problem. Through extensive experiments across a diverse set of tasks, we demonstrated that DiaBlo consistently achieves strong or state-of-the-art performance while maintaining a comparable or smaller number of trainable parameters. Moreover, DiaBlo generalizes well to low-precision settings: in both 4-bit and 2-bit quantized models, it outperforms existing PEFT methods for quantized models without relying on carefully designed quantization or initialization strategies. This robustness and simplicity make DiaBlo an appealing choice for scalable and reliable fine-tuning of large language models. In conclusion, DiaBlo offers a practical and effective approach for efficient model fine-tuning. As AI models continue to scale, the demand for robust and straightforward fine-tuning techniques becomes increasingly important. DiaBlo addresses this demand by achieving a strong balance between performance, efficiency, and implementation simplicity.

## REPRODUCIBILITY

All conducted experiments are reproducible. All models and datasets used are publicly available, as stated in Section 4, and the finetuning hyperparameters are detailed in Appendix A.4. The codes are available at `https://github.com/ziyangjoy/DiaBlo`.

## ACKNOWLEDGMENT

This work was in part supported by NSF grants DMS-2208126, DMS-2110836, IIS-2110546, SUNY-IBM AI Research Alliance grant, UAlbany-IBM CEAIS seed grant, and a start-up grant from UAlbany.

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

# A APPENDIX

## A.1 PROOFS

### A.1.1 PROOF OF THEOREM 1

**Theorem 1.** *Suppose that $\mathbf{X}$ is a generic rank-$r$ matrix. If the number of diagonal blocks $N \leq \frac{m_1}{r}$ is a common factor of $m_1, m_2$, then any solution to the DiaBlo-LSQ (4) also solves the full-LSQ (3).*

*Proof.* Let $d_1 = m_1/N, d_2 = m_2/N$. We write the partitioning $\mathbf{X} = (\mathbf{X}_1 \quad \cdots \quad \mathbf{X}_N)$ and $\mathbf{Z} := \mathbf{Y} - \mathbf{X}\mathbf{W}_0 = (\mathbf{Z}_1 \quad \cdots \quad \mathbf{Z}_N)$, respectively, with $\mathbf{X}_i \in \mathbb{R}^{b \times d_1}$ and $\mathbf{Z}_i \in \mathbb{R}^{b \times d_2}$. Then the DiaBlo-LSQ problem (4) is equivalent to solving

$$\min_{\mathbf{D}_i} \sum_{i=1}^{N} \|\mathbf{Z}_i - \mathbf{X}_i \mathbf{D}_i\|_F^2, \quad i = 1, \ldots, N,$$

each of which is a standard linear least squares problem, and hence has the minimizer

$$\mathbf{D}_i^* = \mathbf{X}_i^\dagger \mathbf{Z}_i, \quad i = 1, \ldots, N, \tag{5}$$

where $\dagger$ denotes the pseudoinverse. Collectively, $\mathbf{D}^* = \text{diag}(\mathbf{D}_1^* \cdots \mathbf{D}_N^*)$ is a solution to the original DiaBlo-LSQ problem (4).

Next, we show that $\mathbf{D}^*$ is also a minimizer of the full-LSQ problem (3). To this end, we rewrite $\mathbf{D}^*$ in the block matrix form

$$\mathbf{D}^* = \begin{pmatrix} \mathbf{D}_{11}^* & \cdots & \mathbf{D}_{1N}^* \\ \vdots & \ddots & \vdots \\ \mathbf{D}_{N1}^* & \cdots & \mathbf{D}_{NN}^* \end{pmatrix},$$

where $\mathbf{D}_{ii}^* := \mathbf{D}_i^*$, and $\mathbf{D}_{ij}^* = \mathbf{0}_{d_1 \times d_2}$ for $i \neq j$. Then for full-LSQ, since $\mathbf{g}_{\mathbf{W}} = -2\mathbf{X}^\top(\mathbf{Z} - \mathbf{X}\mathbf{W})$, it holds that

$$\mathbf{g}_{\mathbf{D}_{ij}^*} = -2\mathbf{X}_i^\top \left( \mathbf{Z}_j - \mathbf{X} \begin{pmatrix} \mathbf{D}_{1j}^* \\ \vdots \\ \mathbf{D}_{Nj}^* \end{pmatrix} \right) = -2\mathbf{X}_i^\top (\mathbf{Z}_j - \mathbf{X}_j \mathbf{D}_{jj}^*) = 2\mathbf{X}_i^\top \mathbf{X}_j \mathbf{D}_j^* - 2\mathbf{X}_i^\top \mathbf{Z}_j.$$

Since the full-LSQ problem (3) is convex, it suffices to show that $\mathbf{g}_{\mathbf{D}_{ij}^*} = \mathbf{0}_{d_1 \times d_2}$ for all $i, j$.

Suppose there holds the low-rank decomposition $\mathbf{X} = \mathbf{R}\mathbf{P}$, where $\mathbf{R} \in \mathbb{R}^{b \times r}$ has orthonormal columns and $\mathbf{P} \in \mathbb{R}^{r \times m_1}$. Alternatively, we assume the partitioning $\mathbf{X}_i = \mathbf{R}\mathbf{P}_i \in \mathbb{R}^{b \times d_1}$, where $\mathbf{P}_i \in \mathbb{R}^{r \times d_1}$. Since $\mathbf{X}$ is a generic rank $r$ matrix, if we choose the number of diagonal blocks $N \leq \frac{m_1}{r}$, or equivalently, $r \leq \frac{m_1}{N} = d_1$, then the matrix $\mathbf{P}_i$ has full row rank of $r$, and thus

$$\mathbf{P}_j \mathbf{P}_j^\dagger = \mathbf{I}_r \tag{6}$$

Moreover, we have

$$\mathbf{X}_j^\dagger = (\mathbf{R}\mathbf{P}_j)^\dagger = \mathbf{P}_j^\dagger \mathbf{R}^\top. \tag{7}$$

Using (5), (6), (7) and that $\mathbf{R}^\top \mathbf{R} = \mathbf{I}_r$, we have

$$\mathbf{X}_i^\top \mathbf{X}_j \mathbf{D}_j^* = \mathbf{X}_i^\top \mathbf{X}_j \mathbf{X}_j^\dagger \mathbf{Z}_j = (\mathbf{P}_i^\top \mathbf{R}^\top)(\mathbf{R}\mathbf{P}_j)(\mathbf{P}_j^\dagger \mathbf{R}^\top)\mathbf{Z}_j = \mathbf{P}_i^\top \mathbf{R}^\top \mathbf{Z}_j = \mathbf{X}_i^\top \mathbf{Z}_j.$$

Therefore, $\mathbf{g}_{\mathbf{D}_{ij}^*} = 2\mathbf{X}_i^\top \mathbf{X}_j \mathbf{D}_j^* - 2\mathbf{X}_i^\top \mathbf{Z}_j = \mathbf{0}_{d_1 \times d_2}$ for all $i, j$. It proves the block diagonal matrix $\mathbf{D}^*$ is a minimizer of full-LSQ (3). Therefore, the DiaBlo-LSQ and full-LSQ have the same minimum value. As a result, every solution to the DiaBlo-LSQ problem is a solution to the full-LSQ problem.

$\square$

A.1.2 PROOF OF THEOREM 2

We first prove the following auxiliary lemma.

**Lemma 1.** *Suppose that $N \geq \lceil \frac{r_1}{d_1} \rceil \lceil \frac{r_2}{d_2} \rceil$. For generic matrices $\{\mathbf{P}_i \in \mathbb{R}^{r_1 \times d_1}\}_{i=1}^N$ and $\{\mathbf{Q}_i \in \mathbb{R}^{r_2 \times d_2}\}_{i=1}^N$, the following matrix*

$$\mathbf{F} = \begin{pmatrix} \mathbf{Q}_1^\top \otimes \mathbf{P}_1^\top \\ \vdots \\ \mathbf{Q}_N^\top \otimes \mathbf{P}_N^\top \end{pmatrix} \in \mathbb{R}^{Nd_1d_2 \times r_1r_2}$$

*has full column rank.*

*Proof.* The conclusion is equivalent to $\det(\mathbf{F}^\top \mathbf{F}) \neq 0$ for generic $\mathbf{P}_i$'s and $\mathbf{Q}_i$'s. Since $\det(\mathbf{F}^\top \mathbf{F})$ is a polynomial in terms of entries in $\mathbf{P}_i$'s and $\mathbf{Q}_i$'s, it suffices to show there exist some $\mathbf{P}_i$'s and $\mathbf{Q}_i$'s such that $\mathbf{F}$ has full column rank (Cox et al., 1997). We construct such matrices in the following.

For convenience, we denote $k_1 = \lceil \frac{r_1}{d_1} \rceil, k_2 = \lceil \frac{r_2}{d_2} \rceil$. Since the $N > k_1k_2$ case can be reduced to $N = k_1k_2$ case, we only consider $N = k_1k_2$. Let $\mathbf{u}_1, \ldots, \mathbf{u}_{r_1}$ be a basis of $\mathbb{R}^{r_1}$ and $\mathbf{v}_1, \ldots, \mathbf{v}_{r_2}$ be a basis of $\mathbb{R}^{r_2}$, then we construct

$$\mathbf{A}_j = (\mathbf{u}_{(j-1)d_1+1}, \ldots, \mathbf{u}_{jd_1}) \in \mathbb{R}^{r_1 \times d_1}, \ 1 \leq j \leq k_1 - 1,$$

$$\mathbf{B}_j = (\mathbf{v}_{(j-1)d_2+1}, \ldots, \mathbf{v}_{jd_2}) \in \mathbb{R}^{r_2 \times d_2}, \ 1 \leq j \leq k_2 - 1,$$

$$\mathbf{A}_{k_1} = (\mathbf{u}_{(k_1-1)d_1+1}, \ldots, \mathbf{u}_{r_1}, \mathbf{0}, \ldots, \mathbf{0}) \in \mathbb{R}^{r_1 \times d_1},$$

$$\mathbf{B}_{k_2} = (\mathbf{v}_{(k_2-1)d_2+1}, \ldots, \mathbf{v}_{r_2}, \mathbf{0}, \ldots, \mathbf{0}) \in \mathbb{R}^{r_2 \times d_2}.$$

The above construction must exist because $k_1 = \lceil \frac{r_1}{d_1} \rceil, k_2 = \lceil \frac{r_2}{d_2} \rceil \Rightarrow k_1d_1 \geq r_1, k_2d_2 \geq r_2$. Let $\mathbf{P}_i = \mathbf{A}_{\lceil i/k_2 \rceil}, \mathbf{Q}_i = \mathbf{B}_{((i-1) \bmod k_2)+1}$ for $i = 1, \ldots, N$, then the rows of matrix $\mathbf{F}$ contain vectors $\{\mathbf{v}_j^\top \otimes \mathbf{u}_i^\top\}_{1 \leq i \leq r_1, 1 \leq j \leq r_2}$, which span the whole $\mathbb{R}^{r_1r_2}$ space. Therefore, $\mathbf{F}$ has rank $r_1r_2$ and hence it has full column rank.

$\square$

**Theorem 2.** *Suppose the activation matrix $\mathbf{X} \in \mathbb{R}^{b \times m_1}$ and the linear output gradient $\mathbf{g}_\mathbf{Y} \in \mathbb{R}^{b \times m_2}$ are generic with ranks $r_1$ and $r_2$, respectively. If $N$ is a common factor of $m_1, m_2$ and satisfies $N \geq \lceil \frac{r_1N}{m_1} \rceil \lceil \frac{r_2N}{m_2} \rceil$, then any adaptation $\mathbf{D} = \mathrm{diag}(\mathbf{D}_1, \ldots, \mathbf{D}_N) \in \mathbb{R}^{m_1 \times m_2}$ produced by DiaBlo that satisfies the stationarity conditions $\mathbf{g}_{\mathbf{D}_i} = \mathbf{0}_{d_1 \times d_2}$ for all $1 \leq i \leq N$ yields adapted weights $\mathbf{W} = \mathbf{W}_0 + \mathbf{D}$ that is also a stationary point of the full FT objective, i.e., $\mathbf{g}_\mathbf{W} = \mathbf{0}_{m_1 \times m_2}$.*

*Proof.* DiaBlo introduces a block-diagonal adaptation $\mathbf{D} = \mathrm{diag}(\mathbf{D}_1, \ldots, \mathbf{D}_N) \in \mathbb{R}^{m_1 \times m_2}$ with each $\mathbf{D}_i \in \mathbb{R}^{d_1 \times d_2}$ (here, $d_1 := m_1/N$ and $d_2 := m_2/N$), and the adapted linear output is given by

$$\mathbf{Y} = \mathbf{X}\mathbf{W} = \mathbf{X}\mathbf{W}_0 + \mathbf{X}\mathbf{D}.$$

Suppose the activations $\mathbf{X} \in \mathbb{R}^{b \times m_1}$ and output $\mathbf{Y} \in \mathbb{R}^{b \times m_2}$ have the corresponding partitioning $\mathbf{X} = (\mathbf{X}_1 \ \cdots \ \mathbf{X}_N)$ and $\mathbf{Y} = (\mathbf{Y}_1 \ \cdots \ \mathbf{Y}_N)$, respectively, with $\mathbf{X}_i \in \mathbb{R}^{b \times d_1}$ and $\mathbf{Y}_i \in \mathbb{R}^{b \times d_2}$. Recall from (2) that the gradient of loss with respect to each tunable block $\mathbf{D}_i$ is

$$\mathbf{g}_{\mathbf{D}_i} = \mathbf{X}_i^\top \mathbf{g}_{\mathbf{Y}_i}, \quad i = 1, \ldots, N,$$

where $\mathbf{g}_{\mathbf{Y}_i} \in \mathbb{R}^{b \times d_2}$ are the $i$-th output gradient block.

When converged, the DiaBlo adaptation $\mathbf{D}$ satisfies the stationarity conditions:

$$\mathbf{g}_{\mathbf{D}_i} = \mathbf{X}_i^\top \mathbf{g}_{\mathbf{Y}_i} = \mathbf{0}_{d_1 \times d_2}, \quad i = 1, \ldots, N. \tag{8}$$

Suppose there holds the low-rank decompositions for the activations $\mathbf{X} = \mathbf{U}\mathbf{P}$ with $\mathbf{U} \in \mathbb{R}^{b \times r_1}$ and $\mathbf{P} \in \mathbb{R}^{r_1 \times m_1}$, and that for the output gradient $\mathbf{g}_\mathbf{Y} = \mathbf{V}\mathbf{Q}$ with $\mathbf{V} \in \mathbb{R}^{b \times r_2}$ and $\mathbf{Q} \in \mathbb{R}^{r_2 \times m_2}$. Alternatively, we assume the partitioning:

$$\mathbf{X}_i = \mathbf{U}\mathbf{P}_i, \quad \mathbf{g}_{\mathbf{Y}_i} = \mathbf{V}\mathbf{Q}_i, \quad i = 1, \ldots, N,$$

where $\mathbf{U}$ and $\mathbf{V}$ span the column spaces of $\mathbf{X}$ and $\mathbf{g_Y}$, respectively, and $\mathbf{P}_i \in \mathbb{R}^{r_1 \times d_1}$, $\mathbf{Q}_i \in \mathbb{R}^{r_2 \times d_2}$. Then the stationarity conditions in (8) gives

$$\mathbf{X}_i^\top \mathbf{g_{Y}}_i = \mathbf{P}_i^\top \mathbf{U}^\top \mathbf{V} \mathbf{Q}_i = \mathbf{0}_{d_1 \times d_2}, \quad i = 1, \dots, N.$$

Using the celebrated Kronecker-vec identity, we have $\mathrm{vec}(\mathbf{P}_i^\top \mathbf{U}^\top \mathbf{V} \mathbf{Q}_i) = (\mathbf{Q}_i^\top \otimes \mathbf{P}_i^\top)\,\mathrm{vec}(\mathbf{U}^\top \mathbf{V})$. So the above stationarity conditions become

$$(\mathbf{Q}_i^\top \otimes \mathbf{P}_i^\top)\,\mathrm{vec}(\mathbf{U}^\top \mathbf{V}) = \mathbf{0}_{d_1 d_2}, \quad i = 1, \dots, N.$$

Stacking all the $N$ equations above leads to the linear system:

$$\mathbf{F}\,\mathrm{vec}(\mathbf{U}^\top \mathbf{V}) = \mathbf{0}_{N d_1 d_2}, \tag{9}$$

where the coefficient matrix is

$$\mathbf{F} = \begin{pmatrix} \mathbf{Q}_1^\top \otimes \mathbf{P}_1^\top \\ \vdots \\ \mathbf{Q}_N^\top \otimes \mathbf{P}_N^\top \end{pmatrix} \in \mathbb{R}^{N d_1 d_2 \times r_1 r_2}.$$

Since the assumption $N \geq \lceil \frac{r_1 N}{m_1} \rceil \lceil \frac{r_2 N}{m_2} \rceil$ is equivalent to $N \geq \lceil \frac{r_1}{d_1} \rceil \lceil \frac{r_2}{d_2} \rceil$, using Lemma 1, $\mathbf{F}$ has full column rank for generic $\mathbf{P}_1, \dots, \mathbf{P}_N, \mathbf{Q}_1, \dots, \mathbf{Q}_N$. So the only solution to (9) is $\mathrm{vec}(\mathbf{U}^\top \mathbf{V}) = \mathbf{0}_{r_1 r_2}$, i.e., $\mathbf{U}^\top \mathbf{V} = \mathbf{0}_{r_1 \times r_2}$. Consequently,

$$\mathbf{g_W} = \mathbf{X}^\top \mathbf{g_Y} = \mathbf{P}^\top \mathbf{U}^\top \mathbf{V} \mathbf{Q} = \mathbf{0}_{m_1 \times m_2}.$$

That is, the full-weight gradient vanishes, including those components with respect to off-diagonal entries of $\mathbf{W}$. Therefore, the stationary point of DiaBlo permits a stationary point of full finetuning when $N \geq \lceil \frac{r_1 N}{m_1} \rceil \lceil \frac{r_2 N}{m_2} \rceil$. $\qquad\square$

## A.2 EMPIRICAL DEMONSTRATION OF THEORETICAL GUARANTEES

### A.2.1 VERIFICATION OF LOW-RANK ASSUMPTION

Theorem 2 relies on the low-rankness of the activation matrix $\mathbf{X}$ and output gradient $\mathbf{g_Y}$. We empirically verify the low-rank assumptions usually hold in large language models.

For a matrix $\mathbf{U} \in \mathbb{R}^{m \times n}$ of rank $r$, let $\sigma_1 \geq \sigma_2 \geq \cdots \geq \sigma_r$ be singular values of $\mathbf{U}$. We define the $p$ effective rank $r_p$ of $\mathbf{U}$ as the smallest number such that $\frac{\sum_{i=1}^{r_p} \sigma_i^2}{\sum_{i=1}^{r} \sigma_i^2} \geq p$. The $p$ effective ratio is $\alpha_p = \frac{r_p}{\min(m,n)}$. We randomly select a decoder from a Llama3-8B model, which is finetuned on the GSM8K dataset, and compute the activation matrices and output gradients. The $95\%$ effective ratios are reported in Figure 3. It is clear that all activation matrices and output gradients have low ranks.

### A.2.2 EMPIRICAL DEMONSTRATION

We use DiaBlo with $N = 64$ to finetune the Llama3-8B model on the GSM8K dataset and record the average gradient norm at the end of each epoch. The average norm of a matrix $\mathbf{U}$ is defined as $\frac{\|\mathbf{U}\|_F}{\sqrt{\#\mathbf{U}}}$, which normalizes by the number of parameters and prevents the full-weight gradients from dominating purely due to scale. This allows for a fair comparison between the gradient signal received by DiaBlo's diagonal blocks and that of the full weight matrix.

We fine-tune only the diagonal blocks while monitoring the gradients of both the full weight matrix and the diagonal blocks. Figure 4 shows the average gradient norms of the two during Llama3-8B fine-tuning on GSM8K. Across all epochs, the curves nearly overlap, indicating that the diagonal blocks receive gradient signals of almost identical magnitude to those of the full model. After the first few epochs, both gradients decay at the same rate and remain tightly aligned. This empirical observation confirms that updating only diagonal blocks is sufficient to drive the full-model gradient toward zero, consistent with the conclusion of our theorem.

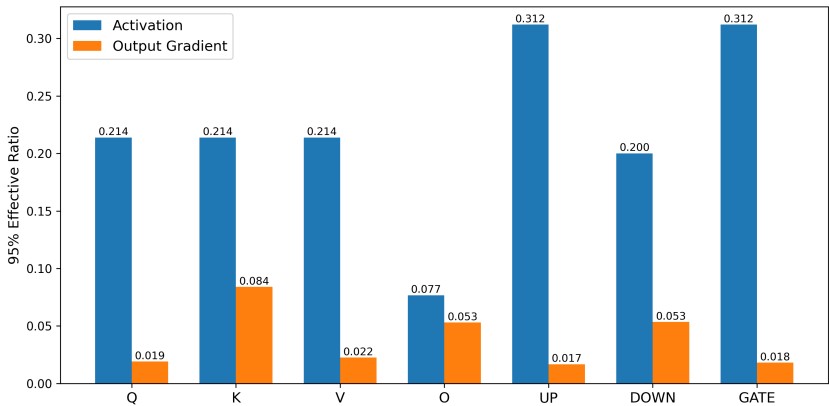

Figure 3: 95% effective rank ratios of modules in finetuned Llama3-8B

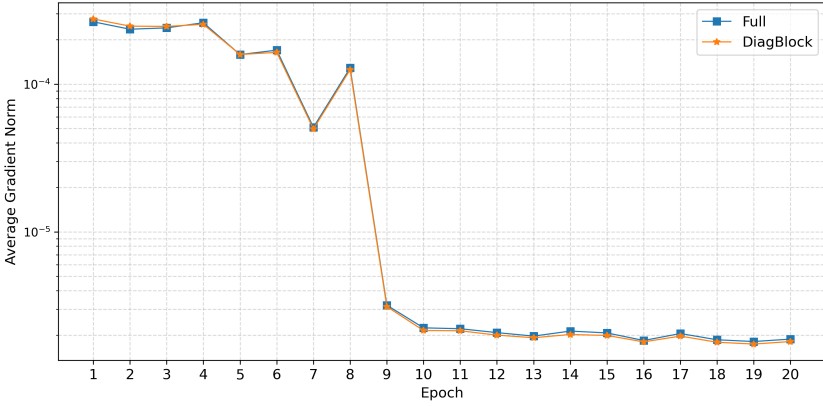

Figure 4: Comparison of average gradient norms for full weights and diagonal-blocks.

### A.3 Additional Experiments

#### A.3.1 Commonsense Reasoning in BF16

The experiments in existing literature, e.g., DoRA (Liu et al., 2024), Pissa (Meng et al., 2024), and MiLoRA (Wang et al., 2024a), all follow the fine-tuning setting in (Hu et al., 2023), which uses `FP16` for model weights, `FP32` for low-rank adaptation, and `FP16` mixed-precision training. Our results reported in Table 1 follow the same fine-tuning setting. However, we notice that LoRA finetuning in precision `BF16` may achieve better results. For comparison, we also report results of finetuning on commonsense reasoning tasks trained in `BF16` precision following the setting in LoRI (Zhang et al., 2025a). The results are reported in Table 6. On **LLaMA3-8B**, DiaBlo achieves the highest average score of 88.3% with $N = 64$ blocks and 1.51% of trainable parameters, surpassing LoRA, DoRA, and LoRI by at least **1.0%**. Even when reducing the number of trainable parameters to 0.76% with $N = 128$ blocks, DiaBlo maintains strong performance, achieving 88.0%, again outperforming the baselines. On **Mistral-7B**, DiaBlo achieves an average accuracy of 88.0% with $N = 64$ blocks, exceeding LoRA, DoRA, and LoRI. At $N = 128$ blocks with less trainable parameters, it continues to perform competitively, reaching 87.9%. These results confirm that DiaBlo is highly effective under `BF16` precision, offering both superior accuracy and parameter efficiency compared to other state-of-the-art PEFT methods.

Table 6: Commonsense reasoning results of fine-tuning Llama3-8B and Mistral-7B in `BF16` following LoRI. Results of baseline methods are taken from LoRI (Zhang et al., 2025a).

| PEFT | r/N | #Params | BoolQ | PIQA | SIQA | HellaS | WinoG | ARC-e | ARC-c | OBQA | AVG |
|------|-----|---------|-------|------|------|--------|-------|-------|-------|------|-----|
| **Mistral-7B** | | | | | | | | | | | |
| Full FT | NA | 100% | 76.6 | 90.6 | 83.5 | 97.0 | 90.0 | 93.4 | 82.9 | 91.5 | 88.2 |
| LoRA | $r = 32$ | 1.25% | 75.2 | 90.1 | 82.9 | 95.1 | 88.1 | 92.0 | 82.9 | 88.7 | 86.9 |
| DoRA | $r = 32$ | 1.25% | 75.8 | 90.4 | 82.9 | 96.3 | 87.9 | 92.6 | 83.3 | 90.6 | 87.5 |
| LoRI | $r = 32$ | 0.63% | 75.9 | 90.6 | 83.0 | 95.9 | 87.4 | 91.9 | 83.6 | 88.4 | 87.1 |
| DiaBlo | $N = 64$ | 1.68% | 76.6 | 90.9 | 83.7 | 96.9 | 88.6 | 93.8 | 83.2 | 90.5 | **88.0** |
| DiaBlo | $N = 128$ | 0.84% | 76.7 | 91.3 | 83.9 | 96.9 | 88.2 | 93.1 | 83.4 | 89.7 | 87.9 |
| **Llama3-8B** | | | | | | | | | | | |
| Full FT | NA | 100% | 76.2 | 90.3 | 84.0 | 96.8 | 88.3 | 93.3 | 84.0 | 90.5 | 87.9 |
| LoRA | $r = 32$ | 1.12% | 76.3 | 89.8 | 82.7 | 95.8 | 88.7 | 91.7 | 83.4 | 88.4 | 87.1 |
| DoRA | $r = 32$ | 1.12% | 75.9 | 89.8 | 82.7 | 95.3 | 88.2 | 93.2 | 83.5 | 87.9 | 87.1 |
| LoRI | $r = 32$ | 0.56% | 76.4 | 89.0 | 82.7 | 95.9 | 87.9 | 93.6 | 84.2 | 88.5 | 87.3 |
| DiaBlo | $N = 64$ | 1.51% | 76.6 | 90.2 | 83.1 | 96.9 | 88.8 | 93.9 | 85.1 | 91.5 | **88.3** |
| DiaBlo | $N = 128$ | 0.76% | 76.2 | 90.5 | 83.7 | 97.0 | 89.0 | 94.0 | 84.7 | 88.9 | 88.0 |

#### A.3.2 Results on MT-Bench

We follow the training setting of LoRA-GA (Wang et al., 2024d) to finetune Llama2-7B on a 52k subset of the WizardLM dataset (Xu et al., 2023) and evaluate the finetuned model on the MT-Bench dataset (Zheng et al., 2023). The quality of the responses is judged by GPT-4 and we report the first turn scores.

As shown in Table 7, DiaBlo achieves the strongest performance on MT-Bench despite using the same parameter budget. With only $0.3\%$ trainable parameters, DiaBlo reaches a score of **6.26**, outperforming LoRA, PiSSA, DoRA, and LoRA-GA. Notably, DiaBlo even surpasses LoRA-GA with a much more parameters (4.7%). These results highlight the effectiveness and parameter efficiency of DiaBlo for tasks with long input-output sequences.

#### A.3.3 Results on GLUE benchmark

We finetune T5-base (Raffel et al., 2020) model on GLUE benchmark (Wang et al., 2018) and evaluate on development sets.

As shown in Table 8, DiaBlo demonstrates consistently strong performance across the GLUE benchmark. Using the same parameter budget as LoRA, DoRA, and PiSSA, DiaBlo achieves the

Table 7: Test results of fine-tuned Llama2-7B on MT-Bench. Results of baseline methods are taken from LoRA-GA (Wang et al., 2024d).

| Method | r/N | #Params | MT-Bench |
|--------|-----|---------|----------|
| LoRA | $r = 8$ | 0.3% | 5.61 |
| PiSSA | $r = 8$ | 0.3% | 5.30 |
| Dora | $r = 8$ | 0.3% | 5.97 |
| LoRA-GA | $r = 8$ | 0.3% | 5.95 |
| LoRA-GA | $r = 128$ | 4.7% | 6.13 |
| DiaBlo | $N = 320$ | 0.3% | **6.26** |

Table 8: Results of finetuning T5-base on GLUE.

| Method | r/N | #Params | MRPC | CoLA | SST-2 | QNLI | MNLI | Avg |
|--------|-----|---------|------|------|-------|------|------|-----|
| Full FT | – | 100% | 84.6 | 80.7 | 94.8 | 93.2 | 86.3 | 87.9 |
| LoRA | $r=8$ | 1.1% | 84.3 | 79.6 | 94.1 | 93.0 | 85.7 | 87.3 |
| DoRA | $r=8$ | 1.1% | 83.9 | 79.1 | 94.4 | 93.1 | 85.3 | 87.2 |
| PiSSA | $r=8$ | 1.1% | 86.8 | 78.5 | 93.7 | 92.9 | 85.3 | 87.4 |
| DiaBlo | $N=64$ | 1.1% | 86.0 | 80.5 | 94.3 | 93.0 | 85.4 | **87.8** |

highest average score of 87.8, outperforming all competing PEFT methods and coming close to full fine-tuning. DiaBlo delivers robust and balanced improvements, indicating that diagonal-block updates offer an efficient and effective alternative to traditional low-rank adaptations for natural language understanding.

### A.3.4 OPTIMIZATION STABILITY

DiaBlo is inherently more stable by avoiding LoRA's matrix-product parameterization. To support this, we measure the gradient norm variance for DiaBlo with $N = 64$ and LoRA with $r = 32$ on LLaMA3-8B over three epochs of commonsense reasoning finetuning. We record the gradient norm every 100 steps and calculate the variance every epoch. As shown in Table 9, DiaBlo's variance consistently decreases, reaching a very low value in the last epoch. It demonstrates that the convergence of DiaBlo is stable and smooth. In contrast, LoRA's variance remains high throughout training; its $A$-matrix variance is artificially low in the first epoch only because its $B$-matrix is initialized to zero. This result suggests that DiaBlo achieves a more stable gradient flow and a more reliable optimization process without special initializations and customized optimizers.

Table 9: Variance of gradient norm for LoRA and DiaBlo on LLaMA3-8B.

| Method | Epoch 1 | Epoch 2 | Epoch 3 |
|--------|---------|---------|---------|
| LoRA-A | 6.13 | 4.97 | 4.48 |
| LoRA-B | 22.34 | 6.75 | 5.63 |
| DiaBlo | 10.17 | **3.10** | **0.75** |

### A.3.5 IMPACT OF HYPERPARAMETER N AND LEARNING RATES

The number of blocks, $N$, controls the trade-off between parameter count and model capacity. We analyze its impact by fine-tuning LLaMA3.2-3B with varying $N$. As shown in Table 10, performance steadily improves as $N$ decreases (more parameters), but the gains diminish and plateau around $N = 64$. This indicates that $N = 64$ provides a strong balance of high performance and parameter efficiency for this model scale.

We further evaluate the performance of DiaBlo under different numbers of diagonal blocks $N$ and learning rates by fine-tuning Llama3-8B on GSM8K and reporting the test accuracy. The results in Figure 5 show that DiaBlo achieves strong accuracy across a wide range of learning rates for all configurations, demonstrating robust optimization behavior. Overall, the performance improves as the number of trainable parameters increases and exhibits a plateau around $N=64$, matching the trend observed in our commonsense reasoning experiments.

Table 10: Ablation on the number of blocks ($N$) for LLaMA3.2-3B on commonsense reasoning. Performance saturates around N=64.

| N | #Params | BoolQ | PiQA | SIQA | HellaS | WinoG | ARC-e | ARC-c | OBQA | AVG |
|---|---|---|---|---|---|---|---|---|---|---|
| Full-FT | NA | 72.42 | 86.18 | 81.27 | 91.96 | 85.24 | 89.44 | 78.24 | 82.80 | 83.44 |
| 8 | 220.50M | 73.88 | 87.00 | 81.27 | 93.62 | 87.21 | 89.90 | 80.29 | 85.60 | **84.85** |
| 16 | 110.25M | 74.10 | 86.13 | 80.96 | 93.17 | 86.58 | 89.98 | 78.50 | 85.60 | 84.38 |
| 32 | 55.13M | 73.94 | 86.45 | 81.32 | 92.95 | 86.50 | 88.85 | 78.84 | 85.40 | 84.28 |
| 64 | 27.56M | 74.62 | 86.83 | 81.32 | 93.22 | 85.71 | 89.94 | 78.24 | 85.60 | 84.44 |
| 128 | 13.78M | 73.94 | 86.07 | 81.53 | 92.79 | 86.11 | 89.86 | 78.24 | 84.80 | 84.17 |
| 256 | 6.89M | 73.00 | 86.02 | 81.06 | 92.45 | 85.65 | 89.06 | 78.50 | 85.60 | 83.92 |
| 512 | 3.45M | 72.35 | 85.36 | 80.60 | 91.58 | 84.85 | 88.85 | 74.32 | 82.40 | 82.54 |
| 1024 | 1.72M | 71.87 | 84.11 | 79.84 | 89.75 | 84.61 | 86.95 | 74.06 | 81.60 | 81.60 |
| 2048 | 1.31M | 68.90 | 82.43 | 79.58 | 87.82 | 82.95 | 84.85 | 67.66 | 78.40 | 79.07 |

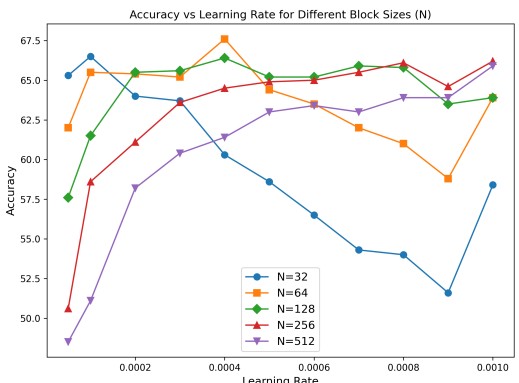

Figure 5: Accuracy of DiaBlo on GSM8K across different learning rates and block $N$.

### A.3.6 FINETUNING SELECTED MODULES

To further evaluate DiaBlo under more restrictive parameter-update settings, we fine-tune Llama3-8B on GSM8K while updating only one module at a time—QK, VUG, or OD—following the module selections used in $S^2$FT (Yang et al., 2024a). The results are shown in Figure 6. For each setting, we choose $N$ and $r$ such that DiaBlo and LoRA use a comparable number of trainable parameters. Across all modules, DiaBlo consistently outperforms LoRA, demonstrating its advantage even when the update space is highly constrained. Moreover, updating the VUG and OD modules yields substantially better performance than updating QK alone, matching the observation reported in $S^2$FT (Yang et al., 2024a).

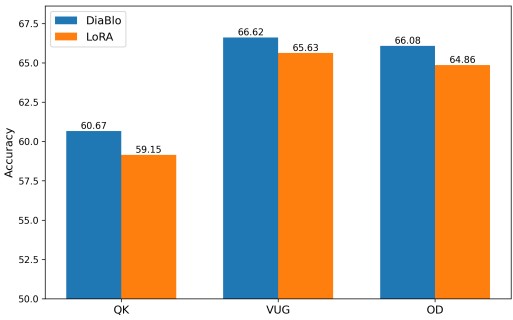

Figure 6: Performance of DiaBlo and LoRA when fine-tuning only selected modules

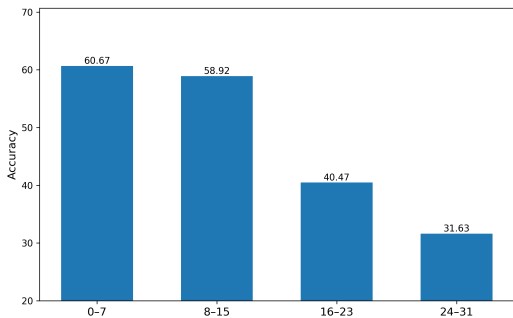

Figure 7: Accuracy on GSM8K when fine-tuning only a contiguous block of layers in Llama3-8B

Table 11: Model Details

| Model name | HuggingFace load name |
| --- | --- |
| Llama-13B | yahma/llama-13b-hf |
| Llama2-7B | meta-llama/Llama-2-7b-hf |
| Llama2-13B | meta-llama/Llama-2-13b-hf |
| Llama3-3B | meta-llama/Llama-3.2-3B |
| Llama3-8B | meta-llama/Meta-Llama-3-8B |
| Mistral-7B | mistralai/Mistral-7B-v0.1 |
| T5-base | t5-base |

### A.3.7 LAYER-WISE SENSITIVITY

To better understand how different layers contribute to DiaBlo's effectiveness, we conduct a layer-wise analysis by fine-tuning only a contiguous block of layers in Llama3-8B and evaluating accuracy on GSM8K. The results are presented in Figure 7.

We observe a clear pattern: fine-tuning early and middle layers yields substantially higher performance than fine-tuning deeper layers alone. Updating layers 0–7 or 8–15 achieves accuracies of 60.67% and 58.92%, respectively, while restricting updates to layers 16–23 leads to a large drop to 40.47%, and layers 24–31 perform the worst at only 31.63%. This trend highlights a strong layer-wise sensitivity in reasoning tasks: early and mid-level representations in LLMs contain far more adaptable structure, whereas the deepest layers are significantly less effective when updated in isolation.

## A.4 IMPLEMENTATION DETAILS

### A.4.1 MODELS

Details of the evaluated models are provided in Table 11. All models used in our experiments are non–instruction-tuned.

### A.4.2 COMMONSENSE REASONING

The results in Table 1 on commonsense reasoning in `FP16` follow the standard setting in (Hu et al., 2023; Liu et al., 2024). The results in Table 6 on commonsense reasoning in `BF16` follow the setting in (Zhang et al., 2025a). The hyper-parameteres for `FP16` are provided in Table 12 and the hyper-parameteres for `BF16` are provided in Table 13.

### A.4.3 ARITHMETIC REASONING

We follow the training setting of MiLoRA (Wang et al., 2024a) for the arithmetic reasoning task. The hyper-parameter setting for fine-tuning Llama2-7B on MetaMathQA dataset is shown in Table 14.

Table 12: Hyper-parameters for the finetuning of Llama2-7B and Llama3-8B on commonsense reasoning in `FP16`.

| Hyper-parameter | Llama2-7B | | Llama3-8B | |
|---|---|---|---|---|
| number of blocks | 64 | 128 | 64 | 128 |
| Optimizer | | AdamW | | |
| Learning rate | 2e-4 | 3e-4 | 5e-5 | 1e-4 |
| Weight decay | | 0 | | |
| LR scheduler | | linear | | |
| Warmup steps | | 100 | | |
| Epochs | | 3 | | |
| Batch size | | 16 | | |
| Max sequence length | | 256 | | |
| Modules | | Q K V U D | | |

Table 13: Hyper-parameters for the finetuning of Mistral-7B and Llama3-8B on commonsense reasoning in `BF16`.

| Hyper-parameter | Mistral-7B | | Llama3-8B | |
|---|---|---|---|---|
| number of blocks | 64 | 128 | 64 | 128 |
| Optimizer | | AdamW | | |
| Learning rate | 5e-5 | 5e-5 | 1e-4 | 2e-4 |
| Weight decay | | 0 | | |
| LR scheduler | | linear | | |
| Warmup steps | | 100 | | |
| Epochs | | 1 | | |
| Batch size | | 32 | | |
| Max sequence length | | 512 | | |
| Modules | | Q K V O G U D | | |

### A.4.4 CODE GENERATION AND SAFETY ALIGNMENT

We follow the training setting of LoRI (Zhang et al., 2025a) for the code generation and safety alignment tasks. The hyper-parameter setting for code generation task is shown in Table 15 and the setting for the safety alignment is show in Table 16.

### A.4.5 QUANTIZED MODELS

We finetune with quantized models on MATH-10K dataset. The fine-tuning hyper-parameters are shown in Table 17.

### A.4.6 MT-BENCH

We finetune Llama2-7B on a 52k subset of WizardLM and evaluate on MT-Bench. The hyper-parameters are shown in Table 18.

### A.4.7 GLUE

We finetune the T5-base model on training datasets of GLUE tasks and evaluate on the development sets. The hyper-parameters are shown in Table 19.

### A.4.8 RETAINING AND FINE-TUNING SPARSE UPDATES

For the results in Table 5, retaining sparse updates uses scales $s = 0.75, 0.75, 0.7$ for sparsity $1/32, 1/64, 1/128$, respectively. The finetuning follows the setting in Table 20.

### A.5 THE USE OF LARGE LANGUAGE MODELS

Parts of the manuscript were refined using large language model tools to improve clarity, grammar, and readability.

Table 14: Hyper-parameters for the finetuning of Llama2-7B on MetaMathQA.

| Hyper-parameter | Llama2-7B | |
|---|---|---|
| number of blocks | 32 | 64 |
| Optimizer | AdamW | |
| Learning rate | 2e-4 | |
| Weight decay | 0 | |
| LR scheduler | linear | |
| Warmup steps | 100 | |
| Epochs | 3 | |
| Batch size | 16 | |
| Max sequence length | 2048 | |
| Modules | Q K V U D | |

Table 15: Hyper-parameters for the finetuning of Mistral-7B and Llama3-8B on CodeAlpaca dataset.

| Hyper-parameter | Mistral-7B | | Llama3-8B | |
|---|---|---|---|---|
| number of blocks | 64 | 128 | 64 | 128 |
| Optimizer | AdamW | | | |
| Learning rate | 1e-4 | 1e-4 | 2e-5 | 3e-5 |
| Weight decay | 0 | | | |
| LR scheduler | linear | | constant | |
| Warmup steps | 100 | | | |
| Epochs | 2 | | | |
| Batch size | 32 | | | |
| Max sequence length | 512 | | | |
| Modules | Q K V O G U D | | | |

Table 16: Hyper-parameters for the finetuning of Mistral-7B and Llama3-8B on Saferpaca dataset.

| Hyper-parameter | Mistral-7B | | Llama3-8B | |
|---|---|---|---|---|
| number of blocks | 64 | 128 | 64 | 128 |
| Optimizer | AdamW | | | |
| Learning rate | 1e-4 | | | |
| Weight decay | 0 | | | |
| LR scheduler | linear | | | |
| Warmup steps | 100 | | | |
| Epochs | 1 | | | |
| Batch size | 32 | | | |
| Max sequence length | 512 | | | |
| Modules | Q K V O G U D | | | |

Table 17: Hyper-parameters for the finetuning of Llama2-7B with quantized models.

| Hyper-parameter | Llama2-7B |
|---|---|
| number of blocks | 64 |
| Optimizer | AdamW |
| Learning rate | 7e-4 |
| Weight decay | 1.0 |
| LR scheduler | linear |
| Warmup ratio | 10% |
| Epochs | 3 |
| Batch size | 16 |
| Max sequence length | 512 |
| Modules | Q K V O G U D |

Table 18: Hyper-parameters for the finetuning of Llama2-7B for MT-Bench evaluation.

| Hyper-parameter | Llama2-7B |
|---|---|
| number of blocks | 320 |
| Optimizer | AdamW |
| Learning rate | 1.5e-3 |
| Weight decay | 0 |
| LR scheduler | linear |
| Warmup steps | 100 |
| Epochs | 1 |
| Batch size | 32 |
| Max sequence length | 1024 |
| Modules | Q K V O G U D |

Table 19: Hyper-parameters for the finetuning of T5-base on GLUE.

| Hyper-parameter | MRPC | COLA | SST-2 | QNLI | MNLI |
|---|---|---|---|---|---|
| number of blocks | | | 64 | | |
| Optimizer | | | AdamW | | |
| Learning rate | 1.4e-2 | 1e-2 | 5e-3 | 4e-3 | 2e-3 |
| Weight decay | | | 0 | | |
| LR scheduler | | | cosine | | |
| Warmup ratio | | | 3% | | |
| Epochs | | | 1 | | |
| Batch size | | | 32 | | |
| Max sequence length | | | 128 | | |
| Modules | | All linear except embeddings and LM head | | | |

Table 20: Hyper-parameters for the finetuning of Llama3-8B on GSM8K.

| Hyper-parameter | Llama3-8B |
|---|---|
| Optimizer | AdamW |
| Learning rate | 4e-4 |
| Weight decay | 0 |
| LR scheduler | linear |
| Warmup steps | 20 |
| Epochs | 2 |
| Batch size | 32 |
| Max sequence length | 512 |
| Modules | Q K V O G U D |

