# OpenReview forum: "DiaBlo: Diagonal Blocks Are Sufficient For Finetuning"
_ICLR.cc/2026/Conference — ICLR 2026 Poster_

### Official Review · Reviewer_yEpk · 2025-10-30

**Soundness:** 2
**Presentation:** 3
**Contribution:** 2
**Rating:** 2
**Confidence:** 4

**Summary:**

- The paper proposes a new PEFT method called DiaBlo, which fine-tunes the weights as $\hat{W} = W + D$, where $W$ are the pretrained weights and $D$ are the residual fine-tuning weights as a block-diagonal matrix.
- $D$ is initialized with all zeros
- The method is evaluated on commonsense reasoning, arithmetic reasoning, code generation tasks and safety alignment tasks.

**Strengths:**

- The use of block diagonal matrix for the residual fine-tuning weights is a experimentally motivated.
- The method is easy to implement and computationally efficient.

**Weaknesses:**

1. The experimental comparisons on the quantization experiments are not correct.
    - The papers experiments use a base model with a different quantization method than the baselines, and the baseline results are taken directly from Table 7 in [1].
    - [1] proposes a new quantization method called AiQ, fine-tunes it using LoRA, and then compares it with other quantization methods fine-tuned using LoRA.
    - However, the experiments in the paper initialize the quantized base model from MagR [2], fine-tunes using DiaBlo, and then compares it with results that use multiple different quantization methods fine-tuned using LoRA.
    - To elaborate, baseline results in Table 4 (taken from Table 7 in [1]) are of the nature "Quantization Method x/y/z + LoRA", ment to compare quantization methods. However, DiaBlo results are of the form "Quantization Method MagR + DiaBlo". Hence, there is no way to tell if the performance difference is due to the quantization method or the fine-tuning method.
    - Hence, the quantization method is a confounder that makes the comparisons invalid.
2. The comparison with baselines in all the tables are taken from multiple sources. Even if the high level settings like precision and hypeparameters are matched, the results are not directly comparable as subtle details in the implementation can lead to different results. For a proper comparison, the baseline methods should be run with the same setup as the proposed method.
    - For example, the baseline results in Table x are taken from 3 sources.
    - When the the baseline methods and DiaBlo are evaluated on the same benchmark but with bf16 precision in Table 6, DiaBlo shows minor improvements over LoRA. As the baseline results are taken from a different work, the improvements could be due to a different training setup.
3. Given that LoRA has had immense practical impact, the contribution of a new PEFT method does not have much impact without a significant advantage other than performance alone. In that sense, the contribution of the paper is limited.

---

## References
[1] "ApiQ: Finetuning of 2-Bit Quantized Large Language Model", Liao et al., EMNLP 2024

[2] "MagR: Weight Magnitude Reduction for Enhancing Post-Training Quantization", Zhang et al., NeurIPS 2024

**Questions:**

1. What is the GPU memory consumed by DiaBlo compared to LoRA/DoRA?

---

> ### Author Response · Authors · 2025-11-22
> **Response to Reviewer yEpk**
>
> Dear Reviewer yEpk:
>
> Thank you so much for your feedback! We reply to your concerns and questions in the following. We hope our responses have successfully addressed your concerns.
>
> ## **Responses to weakness:**
> 1 **Q**: Question on quantization experiments.
>
> **A**: Thank you for raising this concern.
>
> LoftQ and ApiQ are specifically designed for fine-tuning quantized models by jointly choosing quantization matrices and initializing LoRA adapters. In contrast, our setting uses a standard post-training quantization method and directly fine-tunes with DiaBlo, so comparing against LoftQ and ApiQ is appropriate.
>
> For comparisons involving GPTQ-LoRA, we agree that the quantization method may introduce a confounder. To isolate the effect of the fine-tuning method, we additionally evaluate GPTQ-DiaBlo and MagR-LoRA under the same 2-bit quantization setting. In both GPTQ and MagR cases, simply replacing LoRA with DiaBlo yields substantial improvements, showing the significant gains from DiaBlo itself.
>
> We have included the full results and discussions in Section 4.4. The new results on finetuning 2-bit Llama2-13B are summarized below.
>
> Table 2. Fine-tuning 2-bit Llama2-13B
> | Method        | r/N   | #Params | GSM8K | SVAMP | MAWPS | AQuA | AVG  |
> |---------------|-------|---------|-------|-------|-------|------|------|
> | GPTQ-LoRA     | r=64  | 239M    | 31.9  | 49.6  | 82.5  | 23.6 | 46.9 |
> | GPTQ-DiaBlo   | N=64  | 189M    | 41.7  | 58.0  | 87.4  | 23.2 | 52.6 |
> |---------------|-------|---------|-------|-------|-------|------|------|
> | MagR-LoRA     | r=64  | 239M    | 38.6  | 60.5  | 81.1  | 24.8 | 51.2 |
> | MagR-DiaBlo   | N=64  | 189M    | 44.5  | 59.9  | 88.2  | 27.6 | 55.1 |
>
>
> ## **Responses to weakness:**
> 2 **Q**: Question on baselines from different resources.
>
> **A**: Thank you for raising this point.
>
> Within each table, DiaBlo uses exactly the same data processing and training setup, including prompts, sequence length, batch size, precision, etc., as the quoted results within each table. The only change we make is replacing the LoRA adapters with DiaBlo adapters and tuning the learning rate. Citing reported results from existing papers is a common practice in the fine-tuning community. This approach is widely viewed as fair and acceptable, provided that one follows the same training setup as cited results, which is the case in our comparisons.
>
> For Table 1 specifically, we cite DoRA, MiLoRA, and SMT because they all follow the training setup of [1] for commonsense reasoning, a widely used benchmark for evaluating PEFT methods. MiLoRA and SMT themselves also cite DoRA's reported results. Since all methods use the same setting, citing them avoids redundant and expensive training runs.
>
> The smaller performance gap in the bf16 experiments arises because DiaBlo is already extremely close to full fine-tuning under these settings. We have added full fine-tuning results to Table 1 and Table 6, which show that DiaBlo consistently matches or nearly matches full fine-tuning across different settings, demonstrating its robustness and strong performance.
>
>
> [1] "Llm-adapters: An adapter family for parameter-efficient fine-tuning of large language models."
>
> 3 **Q**: Contribution
>
> **A**: Thank you for the comment. We agree that LoRA and other PEFT methods are impactful, but this does not make further work unnecessary. Their wide adoption highlights the need for simpler and more robust alternatives. DiaBlo offers such an option: it avoids complicated initialization and optimization while still achieving strong performance. This shows that the PEFT design space remains open and that improving simplicity, stability, and performance is still valuable.
>
> We would also like to emphasize another contribution of the paper: a new theoretical understanding of why updating only diagonal blocks is effective. In the linear least-squares (LSQ) setting, under a mild low-rank assumption, we show that DiaBlo converges to a global minimizer of the full LSQ problem. Under the same setting and trainable-parameter budget, we further establish that DiaBlo is strictly more expressive than LoRA. Beyond the linear case, we prove that DiaBlo converges to a stationary point of full fine-tuning in general nonlinear models. These theoretical results are summarized in Theorem 1 and Theorem 2 in Section 3.2, and we provide empirical validations in Appendix A.2.
>
>
> ## **Responses to questions:**
>
> **Q**: GPU memory.
>
> **A**: DiaBlo's GPU memory usage is almost identical to LoRA when using the same number of trainable parameters, as discussed in Section 4.5. DoRA uses more memory because it fine-tunes both magnitude and direction. For example, with the same trainable budget (LoRA r=8 and DiaBlo N=64 on T5-base), fine-tuning MNLI with batch size 32, max length 128, and FP32 uses 7.1 GB for both DiaBlo and LoRA, while DoRA requires 10.5 GB.

---

> > ### Comment · Reviewer_yEpk · 2025-11-28
> >
> > Thank you for your rebuttal. Most of my concerns have been addressed, and I will increase my score to 6. However, I don't agree with the authors that comparisons with LoftQ and ApiQ are appropriate in the current form. They would be appropriate if ApiQ+LoRA was replaced with ApiQ+DiaBlo. Differnet quantization methods can change the base model in different ways, hence a model quantized with LoftQ/ApiQ will be different from model quantized by GPTQ/MagR.
> >
> > To be more concrete, consider the optimization objective in ApiQ: $argmin_{Q,A,B} ||XW - X^q(Q + AB)||$, where $A,B$ are the low rank matrices. The comparison would be fair if the objective was $argmin_{Q,D} ||XW - X^q(Q + D)||$, where $D$ is the block diagonal matrix. For LoftQ, I'm not sure if there would be a straightforward way to map the formulation to the block diagonal structure, since LoftQ uses a low rank factorization to obtain the quantized weights themselves. This would indeed be orthogonal to the paper, which is presented as a PEFT technique rather than quantization technique. Hence, I don't understand what the authors are trying to convey in Section 4.4. Is the message of the section "DiaBlo can fine-tune quantized models better than LoRA" or "DiaBlo can be used to simplify quantization pipeline by fine-tuning off the shelf quantized models", or both?

---

> > > ### Author Response · Authors · 2025-11-28
> > >
> > > Thank you so much for your response and for increasing the score! We are happy that most of your concerns have been addressed in our rebuttal.
> > >
> > > For the remaining question, the short answer is that “DiaBlo can fine-tune quantized models better than LoRA.” Section 4.4 does not consider the quantization stage or how to simplify the quantization pipeline. Instead, it aims to demonstrate that DiaBlo can be better than LoRA when fine-tuning quantized models. We understand that both ApiQ and LoftQ use different quantization methods than those used for DiaBlo finetuning, because both ApiQ and LoftQ are specialized to produce quantized weights and LoRA initializations that are tailored for LoRA-based fine-tuning. However, DiaBlo combined with an orthogonal PTQ method, without any specialized quantization or initialization, can reach comparable or better performance. We think this demonstrates that DiaBlo fine-tunes quantized models more effectively and robustly than LoRA. Our additional experimental results on GPTQ-DiaBlo and MagR-LoRA in the previous rebuttal further support this claim. We will revise Section 4.4 accordingly to clarify this point and avoid any confusion.
> > >
> > > Jointly finding the quantized weights and the diagonal blocks, similar to what is done in ApiQ and LoftQ, may further improve results. This is an interesting direction that we may investigate in the future.

---

### Official Review · Reviewer_hx7w · 2025-11-01

**Soundness:** 3
**Presentation:** 3
**Contribution:** 3
**Rating:** 6
**Confidence:** 5

**Summary:**

This paper presents DiaBlo, a param. efficient fine-tuning (PEFT) method for LLMs using the diagonal blocks of weight matrices. One of the key novelties in this work is that DiaBlo does not use extra low-rank matrices multiplied together (like LoRA’s A x B structure) to adapt model weights. Instead, it directly updates selected diagonal blocks within the model’s existing weight matrices, which eliminates the need for initialization or custom optimization strategies. The work covers tasks such as commonsense reasoning, arithmetic reasoning, and code generation, showing that DiaBlo can match or outperform LoRA with comparable memory/efficiency results. Furthermore, the method shows robustness under quantized architectures (4/2-bit).

**Strengths:**

1. By removing the complexity of low-rank structures, this work presents a clear alternative to LoRA-style PEFT. The results show that DiaBlo attains comparable performance without the added overhead of extra trainable matrices, simplifying both tuning and optimization.

2. The evaluation spans diverse supervised fine-tuning tasks -- including code generation, arithmetic reasoning, and commonsense reasoning -- covering a balanced range of short to moderate sequence lengths.

3. The results in Table 5 are particularly convincing, showing that random sparse update patterns (and SMT) underperform compared to DiaBlo. This supports the claim that the structured diagonal-block design is the key driver of its performance advantage.

**Weaknesses:**

1. Most evaluated benchmarks involve short output sequences, except for code generation. Testing DiaBlo on tasks with longer input–output contexts would better demonstrate its scalability and performance stability under extended sequence conditions (see q1).

2. The discussion of sparsity-based PEFT methods misses some recent relevant work, such as S2FT (NeurIPS 2025)[1] and SparseLoRA (ICML 2025)[2]. Including these would strengthen the discussion on sparsity in the introduction and would provide a better picture of the current limitations in state-of-the-art PEFT methods.

References:

[1] Xinyu Yang, Jixuan Leng, Geyang Guo, Jiawei Zhao, Ryumei Nakada, Linjun Zhang, Huaxiu Yao, Beidi Chen, "S2FT: Efficient, Scalable and Generalizable LLM Fine-tuning by Structured Sparsity", NeurIPS 2025

[2] Samir Khaki, Xiuyu Li, Junxian Guo, Ligeng Zhu, Chenfeng Xu, Konstantinos N. Plataniotis, Amir Yazdanbakhsh, Kurt Keutzer, Song Han, Zhijian Liu, "SparseLoRA: Accelerating LLM Fine-Tuning with Contextual Sparsity", ICML 2025

**Questions:**

1. Most evaluated tasks focus on short output sequences, such as commonsense and reasoning benchmarks. Including long-form, multi-turn dialogue datasets like MT-Bench would better demonstrate DiaBlo’s scalability and effectiveness in extended conversational contexts.

2. The Appendix shows that trainable modules (QKVO/GUD) are fixed across methods, which is reasonable, but it remains unclear whether DiaBlo’s robustness comes from having a larger set of modules trainable on all methods (for example, for LLaMA3-8B, all the QKVOGUD modules have some trainable parameters). An ablation over different subsets of trainable components would clarify if the observed gains persist under more constrained settings. For example, see Figure 4.0 in S2FT on adding trainable parameters to only a subset of modules, such as QK, or GUD, etc.

3. Adding results on the GLUE benchmark would help assess general language understanding and show how well DiaBlo transfers to broader NLP tasks.

4. Reporting zero-shot baseline performance would contextualize fine-tuning improvements. Clarifying whether the LLaMA3-8B variant used is “instruct” or base would also make the evaluation setup more transparent.

---

> ### Author Response · Authors · 2025-11-22
> **Response to Reviewer hx7w**
>
> Dear Reviewer hx7w:
>
> Thank you so much for your feedback! We reply to your concerns and questions in the following. We hope our responses have successfully addressed your concerns.
>
> In addition, we develop new **theoretical justifications** for DiaBlo in Section 3.2 of the revised paper, which is briefly discussed at the end of this official comment.
>
> ## **Responses to weakness:**
>
> 1 **Q**: Test on longer input-output contexts.
>
> **A**: Thank you for the suggestion!
>
> To evaluate DiaBlo on tasks with longer input-output sequences, we follow the setting of the LoRA-GA paper to fine-tune DiaBlo on the 52k subset of the WizardLM dataset and then evaluate using MT-Bench. With only 0.3% trainable parameters, DiaBlo outperforms all baseline methods. Even when LoRA-GA increases its trainable parameter budget to 4.7%, its performance remains slightly below that of DiaBlo. These results demonstrate DiaBlo's strong performance on longer sequences. The results are shown in the following Table 1 and Appendix A.3.2 of the revised paper.
>
> Table 1. Test results of fine-tuned Llama2-7B on MT-Bench
> | Method   | r / N   | #Params | MT-Bench |
> |----------|---------|---------|----------|
> | LoRA     | r=8     | 0.3%    | 5.61     |
> | PiSSA    | r=8     | 0.3%    | 5.30     |
> | DoRA     | r=8     | 0.3%    | 5.97     |
> | LoRA-GA  | r=8     | 0.3%    | 5.95     |
> | LoRA-GA  | r=128   | 4.7%    | 6.13     |
> | DiaBlo | N=320 | 0.3% | **6.26** |
>
> 2 **Q**: Add citations.
>
> **A**: Thank you so much for pointing out the two interesting works. We have cited the paper and included them in our introduction.
>
> ## **Responses to questions:**
>
> 1 **Q**: Test on longer input-output contexts.
>
> **A**: See response to weakness 1.
>
> 2 **Q**: Finetuning on selected modules
>
> **A**: Thanks for the question! We conducted ablation studies to investigate the importance of different modules. Following the observation from S2FT [1] that OD > VUG >> QK, our results show a similar pattern: fine-tuning OD and VUG yields substantially better performance than fine-tuning QK. Moreover, DiaBlo achieves performance gains over LoRA in the restrictive finetuning setting. The results are presented in the following Table 2 and Appendix A.3.6 of the revised paper.
>
> Table 2. Performance of DiaBlo and LoRA when fine-tuning only selected modules
> | Method | QK    | VUG   | OD    |
> |--------|-------|-------|-------|
> | DiaBlo | 60.67 | 66.62 | 66.08 |
> | LoRA   | 59.15 | 65.63 | 64.86 |
>
> 3 **Q**: Glue benchmark.
>
> **A**: Thanks for the suggestion! We evaluated DiaBlo on the GLUE benchmark using the T5-base model, and the results are presented in Table 3 below. With the same trainable-parameter budget, DiaBlo achieves slightly better average performance than the baseline methods. We have included the full details and discussion in Appendix A.3.3 of the revised paper.
>
> Table 3. Results of finetuning T5-base on GLUE.
>
> | Method   | r/N     | #Params | MRPC | CoLA | SST-2 | QNLI | MNLI | Avg   |
> |----------|---------|---------|------|------|-------|------|------|-------|
> | Full FT  | --      | 100%    | 84.6 | 80.7 | 94.8  | 93.2 | 86.3 | 87.9  |
> | LoRA     | r=8     | 1.1%    | 84.3 | 79.6 | 94.1  | 93.0 | 85.7 | 87.3  |
> | DoRA     | r=8     | 1.1%    | 83.9 | 79.1 | 94.4  | 93.1 | 85.3 | 87.2  |
> | PiSSA    | r=8     | 1.1%    | 86.8 | 78.5 | 93.7  | 92.9 | 85.3 | 87.4  |
> | DiaBlo   | N=64    | 1.1%    | 86.0 | 80.5 | 94.3  | 93.0 | 85.4 | **87.8** |
>
>
> 4 **Q**: Zero-shot results and model detail.
>
> **A**: Thanks for the suggestion! We have included the zero-shot results in Table 1, Table 2, and Table 3. The zero-shot results are much worse than the results after finetuning.
>
> Regarding the model specification, we use the base Llama3-8B model (HuggingFace name: meta-llama/Meta-Llama-3-8B). We have added Appendix A.4.1 and Table 11 to provide detailed descriptions of all models used in the paper.
>
>
> ## **Theoretical Justification**
> In addition to addressing your concerns, we would also like to draw your attention to another major update in the paper: we now provide new theoretical justifications for the effectiveness of updating only diagonal blocks in DiaBlo. In the linear least-squares (LSQ) problem, under a mild low-rank assumption, we show that DiaBlo converges to a global minimizer of the full LSQ problem. Under the same setting and trainable-parameter budget, we further establish that DiaBlo is strictly more expressive than LoRA. Beyond the linear case, we prove that DiaBlo converges to a stationary point of full fine-tuning in general nonlinear models under similar low-rank assumption. The theoretical results partically explain why DiaBlo is effective in finetuning. The results are summarized in Theorem 1 and Theorem 2 in Section 3.2, and we provide empirical validations in Appendix A.2.

---

### Official Review · Reviewer_qumS · 2025-11-01

**Soundness:** 2
**Presentation:** 3
**Contribution:** 2
**Rating:** 6
**Confidence:** 3

**Summary:**

This work presents a Parameter-Efficient Fine-Tuning method called DiaBlo that updates only the diagonal blocks of selected model weight matrices. The work argues that this method enables axis-aligned, per-dimension scaling that LoRA cannot capture and verifies the method on a large variety of benchmarks with Llama 7B/8B/13B models.

**Strengths:**

- The proposed work eliminates the inherent optimization difficulties associated with low-rank decomposition by avoiding the use of matrix products.

- DiaBlo demonstrates higher stability in 4-bit and 2-bit arithmetic reasoning tasks.

**Weaknesses:**

- Compared to strong baselines like SMT with similar trainable parameter amount, the proposed method does not show significantly better performance. In other words, the paper argues the memory and computation efficiency of the proposed method, but the model doesn’t achieve significant improved performance compared to baselines when they share the same amount of trainable parameters.

- In table 1, it shows DiaBlo N =128 doesn’t get better performance compared to DiaBlo N =64 although doubled trainable parameters. This raises concerns of the scaling ability of the proposed Diablo.

**Questions:**

- The reviewer suggests to add Full Finetuning results in Table 1.

- The paper mentions when N is not a common factor of m1,m2, it needs to expand and pad the weight into a proper size and then select the corresponding diagonal blocks. This can be common case. But the paper doesn’t touch this point clearly in later experiments.

---

> ### Author Response · Authors · 2025-11-22
> **Response to Reviewer qumS**
>
> Dear Reviewer qumS:
>
> Thank you so much for your feedback! We reply to your concerns and questions in the following. We hope our responses have successfully addressed your concerns.
>
> In addition, we develop new **theoretical justifications** for DiaBlo in Section 3.2 of the revised paper, which is briefly discussed at the end of this official comment.
>
> ## **Responses to weakness:**
> 1 **Q**: Performance comparison.
>
> **A**: Thanks for the comment!
>
> DiaBlo generally achieves comparable or better performance than strong baselines such as SMT, while using the same or fewer trainable parameters.
>
> As shown in Table 1 of the paper, on Llama2-7B for commonsense reasoning, DiaBlo reaches an average accuracy of 83.5% with only 0.52% trainable parameters, compared to 81.8% for SMT with 0.84% parameters. Even when SMT increases its trainable budget to 4.91%, its accuracy (83.4%) remains slightly below DiaBlo's performance achieved with far fewer parameters. A similar pattern appears on Llama3-8B. SMT requires 3.2% trainable parameters to match the performance of DiaBlo with only 0.52%. In addition, DiaBlo already approaches these upper bounds. Furthermore, Tables 2, 3, and 4 provide additional evidence that DiaBlo often matches or outperforms other methods under the same trainable-parameter budget.
>
> | Model        | Method       | Params  | AVG  |
> |--------------|--------------|---------|------|
> | **Llama2-7B** | Full FT      | 100%    | 83.5 |
> |              | SMT          | 0.84%   | 81.8 |
> |              | SMT (Best)   | 4.91%   | 83.4 |
> |              | DiaBlo (N=64)  | 1.04%   | 83.4 |
> |              | DiaBlo (N=128) | 0.52%   | **83.5** |
> |--------------|--------------|---------|------|
> | **Llama3-8B** | Full FT      | 100%    | 87.5 |
> |              | SMT          | 0.71%   | 86.8 |
> |              | SMT (Best)   | 3.01%   | 87.2 |
> |              | DiaBlo (N=64)  | 1.04%   | **87.3** |
> |              | DiaBlo (N=128) | 0.52%   | 87.2 |
>
> 2 **Q**: scaling ability of Diablo.
>
> **A**: Thanks for the comment. We are sorry for any confusion. In our notation, N refers to the number of diagonal blocks, so the number of trainable parameters is actually inversely proportional to N. This means N = 64 has twice as many trainable parameters as N = 128.
>
> Regarding Table 1, the full fine-tuning accuracies for Llama2-7B and Llama3-8B are 83.5% and 87.5%, respectively. DiaBlo with N = 64 achieves 83.4% and 87.3%, and DiaBlo with N = 128 achieves 83.5% and 87.2%. Since both settings already match or are extremely close to full fine-tuning, increasing the trainable parameters does not lead to noticeable gains.
>
> It is clearer to see the scaling trend on more challenging tasks. In Table 2 on hard MATH test dataset, N = 32 (more parameters) significantly outperforms N = 64 (fewer parameters). Appendix A.3.5 also provides an ablation on different block sizes, showing that performance generally improves as the number of trainable parameters increases and then reaches a plateau, where the results already match those of full fine-tuning.
>
>
> ## **Responses to questions.**
>
> 1 **Q**:Full Finetuning results in Table 1.
>
> **A**: Thanks for the suggestion! We have included full finetuning results in Table 1. Our DiaBlo results are very close to full-finetuning results.
>
> 2 **Q**: When N is not a common factor of m1,m2.
>
> **A**: Thank you for the suggestion! We have included more details in Section 3.1.
>
> When $N$ is not a common factor of $m_1,m_2$. In this case, we set $d_1 = \left\lceil \frac{m_1}{N} \right\rceil, d_2 = \left\lceil \frac{m_2}{N} \right\rceil.$ At inference time, we pad ${X}$ with zeros so that its dimension becomes $b \times N d_1$, perform the blockwise computation, and then truncate the resulting output back to the size $b \times m_2$. This situation is uncommon in practice because modern LLM architectures purposely choose hidden sizes and intermediate dimensions that are highly composite ensuring many convenient common factors. The commonly used N=32,64,128 are common factors for most modern LLMs.
>
>
>
> ## **Theoretical Justification**
> In addition to addressing your concerns, we would also like to draw your attention to another major update in the paper: we now provide new theoretical justifications for the effectiveness of updating only diagonal blocks in DiaBlo. In the linear least-squares (LSQ) problem, under a mild low-rank assumption, we show that DiaBlo converges to a global minimizer of the full LSQ problem. Under the same setting and trainable-parameter budget, we further establish that DiaBlo is strictly more expressive than LoRA. Beyond the linear case, we prove that DiaBlo converges to a stationary point of full fine-tuning in general nonlinear models under similar low-rank assumption. The theoretical results partically explain why DiaBlo is effective in finetuning. The results are summarized in Theorem 1 and Theorem 2 in Section 3.2, and we provide empirical validations in Appendix A.2.

---

### Official Review · Reviewer_uDXN · 2025-11-04

**Soundness:** 4
**Presentation:** 3
**Contribution:** 4
**Rating:** 8
**Confidence:** 5

**Summary:**

This paper introduces a new parameter-efficient finetuning method which targets trains a block-diagonal sparse matrix on top of linear layers in model weights. This approach seems to outperform various other PEFTs with around the same number of parameters on a variety of tasks, as well as when compared to other baselines which involve training sparse weight updates.

**Strengths:**

- The idea is quite elegant, relatively simple to implement and efficient to train -- there isn't much adaptation required to existing finetuning libraries to get this working.
- The results are broad (covering standard PEFT benchmarks) and thus convincing.
- Ablations cover the first questions I had regarding whether block-diagonal is better than other ways of selecting entries to tune in the weight matrix; it does seem like it is broadly a better strategy than other ideas.

**Weaknesses:**

- Since we use a standard suite of benchmarks to evaluate PEFTs, it's possible that our literature is engaging in test-set overfitting (compare how the ImageNet challenge or LMSYS arena were overfit by organisations submitting many models repeatedly). It would thus be nice to show how the technique performs under varying learning rates and block sizes (e.g. as done for LoRA in [Schulman et al. (2025)](https://thinkingmachines.ai/blog/lora/)). It is nice though that there are not as many hyperparameters as other PEFTs!

**Questions:**

- In your experience how difficult was it to hyperparameter tune this method?
- What model components seem most important for good performance when finetuning? Is there a layer-wise effect?

---

> ### Author Response · Authors · 2025-11-22
> **Response to Reviewer uDXN**
>
> Dear Reviewer uDXN:
>
> Thank you so much for your feedback! We reply to your concerns and questions in the following. We hope our responses have successfully addressed your concerns.
>
> In addition, we develop new **theoretical justifications** for DiaBlo in Section 3.2 of the revised paper, which is briefly discussed at the end of this official comment.
>
> ## **Responses to weakness:**
> 1 **Q**: Performance under varying learning rates and block sizes.
>
> **A**: Thanks a lot for the suggestion. We conducted additional experiments varying both the learning rate and the number of diagonal blocks $N$ on GSM8K in addition to our previous experimental results on commonsense reasoning with varying block sizes. We observe that as $N$ decreases (i.e., block sizes increase and more parameters are trainable), performance improves and reaches a plateau around $N=64$, where results are already close to full fine-tuning.  The finetuning with more trainable parameters tend to require lower learning rates. The results show that DiaBlo remains stable and effective across a wide range of block sizes and learning rates. We have included the detailed results and discussions in Appendix A.3.5. Table 1 summarizes the commonsense results, while the new GSM8K results are provided in Appendix A.3.5 because of space limitations.
>
>
> Table 1. Results of finetuning Llama3.2-3B on commonsensereasoning with varying block sizes.
> | N    | #Params | AVG   |
> |------|---------|-------|
> | Full-FT   |  | 83.44 |
> | 8    | 220.50M | **84.85** |
> | 16   | 110.25M | 84.38 |
> | 32   | 55.13M  | 84.28 |
> | 64   | 27.56M  | _84.44_ |
> | 128  | 13.78M  | 84.17 |
> | 256  | 6.89M   | 83.92 |
> | 512  | 3.45M   | 82.54 |
> | 1024 | 1.72M   | 81.60 |
> | 2048 | 1.31M   | 79.07 |
>
>
>
> ## **Responses to questions:**
> 1 **Q**: How difficult was it to hyperparameter tune this method?
>
> **A**: Hyperparameter tuning is generally straightforward for our method. Finetuning with DiaBlo primarily involves two hyperparameters: the number of blocks $N$ and the learning rate $lr$. In our experiments, $N \in \{32, 64, 128\}$ generally performs well. The optimal learning rates typically fall within the range of \\(5\times10^{-5}\\) to \\(5\times10^{-4}\\), and a simple sweep over this interval is usually sufficient.
>
>
>
>
> 2 **Q**: What model components seem most important for good performance when finetuning? Is there a layer-wise effect?
>
> **A** Thanks for the question! We conducted ablation studies to investigate the importance of different components and layers. Following the observation from S2FT [1] that OD > VUG >> QK, our results show a similar pattern. Under the similar parameter budget, fine-tuning OD and VUG yields substantially better performance than fine-tuning QK. Moreover, DiaBlo achieves clear performance gains over LoRA in the restrictive finetuning setting. The detailed results are presented in the following Table 2 and Appendix A.3.6 of the revised paper.
>
> Regarding layer-wise effects, we find that fine-tuning bottom layers (close to inputs) is significantly more effective than fine-tuning top layers (close to outputs). The full results are reported in the following Table 3 and Appendix A.3.7 of the revised paper.
>
> Table 2. Performance of DiaBlo and LoRA when fine-tuning only selected modules
> | Method | QK    | VUG   | OD    |
> |--------|-------|-------|-------|
> | DiaBlo | 60.67 | 66.62 | 66.08 |
> | LoRA   | 59.15 | 65.63 | 64.86 |
>
> Table 3. Accuracy on GSM8K when fine-tuning only a contiguous block of layers in Llama3-8B
> | Layer Range | Accuracy |
> |-------------|--------|
> | 0-7         | 60.67  |
> | 8-15        | 58.92  |
> | 16-23       | 40.47  |
> | 24-31       | 31.63  |
>
> [1] "S $^{2} $ FT: Efficient, scalable and generalizable LLM fine-tuning by structured sparsity." Advances in Neural Information Processing Systems
>
> ## **Theoretical Justification**
> In addition to addressing your concerns, we would also like to draw your attention to another major update in the paper: we now provide new theoretical justifications for the effectiveness of updating only diagonal blocks in DiaBlo. In the linear least-squares (LSQ) problem, under a mild low-rank assumption, we show that DiaBlo converges to a global minimizer of the full LSQ problem. Under the same setting and trainable-parameter budget, we further establish that DiaBlo is strictly more expressive than LoRA. Beyond the linear case, we prove that DiaBlo converges to a stationary point of full fine-tuning in general nonlinear models under similar low-rank assumption. The theoretical results partically explain why DiaBlo is effective in finetuning. The results are summarized in Theorem 1 and Theorem 2 in Section 3.2, and we provide empirical validations in Appendix A.2.

---

### Author Response · Authors · 2025-11-22
**Summary of revisions**

Dear reviewers:

Thank you very much for the valuable comments, which have helped us greatly improve the paper. We have revised the manuscript accordingly and marked all major changes in red. The main revisions are summarized below.

1. We develop new theoretical justifications for the effectiveness of updating only diagonal blocks in DiaBlo. In the linear least-squares (LSQ) setting, under a mild low-rank assumption, we show **DiaBlo converges to a global minimizer of the full LSQ problem**. In the same setting, we further demonstrate that, for an equivalent trainable-parameter budget, **DiaBlo is strictly more expressive than LoRA**. We further prove that **DiaBlo converges to a stationary point of full fine-tuning** in general nonlinear models. The theoretical results are summarized in Theorem 1 and Theorem 2 in Section 3.2.

2. For experiments on fine-tuning quantized models, to isolate the effect of quantization, we also evaluate GPTQ-DiaBlo and MagR-LoRA under the 2-bit setting. With both GPTQ quantization and MagR quantization, simply replacing LoRA with DiaBlo significantly improves the results. It indicates that DiaBlo itself drives the performance improvement, consistently enhancing accuracy under low-bit quantization. Moreover, DiaBlo achieves these gains without requiring specialized quantization procedures or initialization strategies.

Table 1. Fine-tuning 2-bit Llama2-7B
| Method        | r/N   | #Params | GSM8K | SVAMP | MAWPS | AQuA | AVG  |
|---------------|-------|---------|-------|-------|-------|------|------|
| GPTQ-LoRA     | r=64  | 112M    | 21.7  | 39.0  | 76.6  | 22.1 | 39.9 |
| GPTQ-DiaBlo   | N=64  | 70M     | 33.3  | 50.9  | 84.0  | 22.8 | 47.8 |
|---------------|-------|---------|-------|-------|-------|------|------|
| MagR-LoRA     | r=64  | 112M    | 30.9  | 46.9  | 86.6  | 20.5 | 46.2 |
| MagR-DiaBlo   | N=64  | 70M     | 32.1  | 51.5  | 87.0  | 24.0 | 48.7 |


Table 2. Fine-tuning 2-bit Llama2-13B
| Method        | r/N   | #Params | GSM8K | SVAMP | MAWPS | AQuA | AVG  |
|---------------|-------|---------|-------|-------|-------|------|------|
| GPTQ-LoRA     | r=64  | 239M    | 31.9  | 49.6  | 82.5  | 23.6 | 46.9 |
| GPTQ-DiaBlo   | N=64  | 189M    | 41.7  | 58.0  | 87.4  | 23.2 | 52.6 |
|---------------|-------|---------|-------|-------|-------|------|------|
| MagR-LoRA     | r=64  | 239M    | 38.6  | 60.5  | 81.1  | 24.8 | 51.2 |
| MagR-DiaBlo   | N=64  | 189M    | 44.5  | 59.9  | 88.2  | 27.6 | 55.1 |



3. We conduct additional experiments on MT-Bench and the GLUE benchmark, along with ablation studies examining the effects of varying block sizes, learning rates, target modules, and layer selections. The corresponding results are reported in Appendices A.3.2, A.3.3, A.3.5, A.3.6, and A.3.7. Together, these studies further validate the robustness and effectiveness of DiaBlo across a broad range of configurations.

---

> ### Author Response · Authors · 2025-11-26
> **Gentle follow-up on our rebuttal (Paper DiaBlo: Diagonal Blocks Are Sufficient For Finetuning)**
>
> Dear Reviewers,
>
> We hope you are doing well. We have posted our rebuttal on OpenReview and have answered the questions and concerns raised in the reviews. New theoretical results for DiaBlo have also been added to the revision. We just wanted to kindly follow up to see if our responses have addressed your concerns, and in case you need anything further from us. We truly appreciate your time and efforts during this busy stage of the review process.
>
> Thank you very much!
>
> Best,
> Authors

---

### Meta-Review · Area_Chair_fo7g · 2026-01-04

**Summary:**

This paper introduces a novel parameter-efficient fine-tuning (PEFT) method that learns a block-diagonal sparse matrix as an update to linear layer weights. The proposed approach outperforms other PEFT methods with comparable parameter budgets across a range of tasks, and also surpasses alternative baselines that train sparse weight updates. The reviewers praised the paper's elegant concept and its strong, broad-ranging experimental results.

**Reviewer Concerns:**

The reviewers asked for clarifications and extra tests to check how well the method works across different tasks and settings. The authors ran many new experiments, including on more datasets and with different model setups. They also added new theory showing why their method is stronger than LoRA in certain cases and matches full fine-tuning in others.

**Reviewer Scores:**

This paper received four reviews, initially yielding three accept and one reject recommendations. During the discussion phase, the dissenting reviewer revised their assessment, resulting in a unanimous consensus for acceptance. Accordingly, the AC recommends accepting the paper.

---

### Decision · Program_Chairs · 2026-01-26

Accept (Poster)